# Type 1 piliated uropathogenic *Escherichia coli* hijack the host immune response by binding to CD14

Kathrin Tomasek[1]*[†], Alexander Leithner[1][‡], Ivana Glatzova[1][§], Michael S Lukesch[2], Calin C Guet[1]*, Michael Sixt[1]*

[1]Institute of Science and Technology Austria, Klosterneuburg, Austria; [2]VALANX Biotech GmbH, Klosterneuburg, Austria

**\*For correspondence:**
kathrin.tomasek@epfl.ch (KT);
calin.guet@ist.ac.at (CCG);
sixt@ist.ac.at (MS)

**Present address:** [†]Laboratory of Microbiology and Microtechnology, Ecole Polytechnique Fédérale de Lausanne (EPFL), Lausanne, Switzerland; [‡]Kennedy Institute of Rheumatology, University of Oxford, Oxford, United Kingdom; [§]University of Copenhagen, Copenhagen, Denmark

**Abstract** A key attribute of persistent or recurring bacterial infections is the ability of the pathogen to evade the host's immune response. Many *Enterobacteriaceae* express type 1 pili, a pre-adapted virulence trait, to invade host epithelial cells and establish persistent infections. However, the molecular mechanisms and strategies by which bacteria actively circumvent the immune response of the host remain poorly understood. Here, we identified CD14, the major co-receptor for lipopolysaccharide detection, on mouse dendritic cells (DCs) as a binding partner of FimH, the protein located at the tip of the type 1 pilus of *Escherichia coli*. The FimH amino acids involved in CD14 binding are highly conserved across pathogenic and non-pathogenic strains. Binding of the pathogenic strain CFT073 to CD14 reduced DC migration by overactivation of integrins and blunted expression of co-stimulatory molecules by overactivating the NFAT (nuclear factor of activated T-cells) pathway, both rate-limiting factors of T cell activation. This response was binary at the single-cell level, but averaged in larger populations exposed to both piliated and non-piliated pathogens, presumably via the exchange of immunomodulatory cytokines. While defining an active molecular mechanism of immune evasion by pathogens, the interaction between FimH and CD14 represents a potential target to interfere with persistent and recurrent infections, such as urinary tract infections or Crohn's disease.

## Editor's evaluation

Sixt and colleagues demonstrate that a protein that sits at the tip of the pilus of a gram-negative bacterium exerts immune evasion functions by binding CD14, the co-receptor of TLR4. The identification of this pathway may help to develop new therapeutic approaches to intervene against recurrent infections and/or inflammatory diseases of the gut.

## Introduction

Cells of metazoan organisms constantly interact with a wide diversity of bacterial species that populate the host organism internally as well as externally. Therefore, the host immune response is faced with a non-trivial problem – accommodate beneficial commensals while at the same time remove harmful pathogens (*Hooper and Macpherson, 2005*). The difficulty of this task lies in the fact that most of the molecular patterns, such as lipopolysaccharide (LPS) or surface organelles such as pili and flagella, are conserved between commensal and pathogenic bacteria. The main difference between commensal and pathogenic strains is the ability of the latter to hijack host cell functions for their own benefit (*Magalhaes et al., 2007*). Consequently, the host immune system uses complex discrimination strategies, such as spatial compartmentalization of the receptors recognizing pathogen signatures

and concurrent sensing of molecular patterns associated with host damage. The discrimination ability of the host is not always perfect, as pathogens can persist asymptomatically in the host for long periods of time or cause symptomatic acute or chronic infections in some individuals, but not in others (*Grant and Hung, 2013*). For example, the commensal bacterium *Escherichia coli*, one of the main residents of the mammalian intestine, occasionally causes clinical infections, especially when the bacteria acquire virulence traits and thus manage to populate extraintestinal host niches (*Magalhaes et al., 2007*; *Leimbach et al., 2013*). Usually disease progression does not pose a dead end to such opportunistic pathogens, since bacteria are shed in high numbers during the infection, allowing the pathogen to populate new environments, a new host, or even recolonize their original host niche – the intestine (*Donnenberg, 2013*).

In principle, any commensal *E. coli* has the potential to evolve into a pathogen by acquiring virulence traits (*Wirth et al., 2006*). Those traits can be either adapted, namely evolved specifically to increase the fitness of a pathogen during the infection, or pre-adapted, meaning even though they increase the fitness of the pathogen, they were originally evolved for a non-virulent function (*Donnenberg, 2013*). Examples of adapted traits are the acquisition of pathogenicity islands through horizontal gene transfer (*Oelschlaeger et al., 2002*) or pathoadaptive mutations, such as changes in the LPS, flagellum or pili components (*Weissman et al., 2006*; *Donnenberg, 2013*). A prime example of a pre-adapted trait are type 1 pili. Both commensal and pathogenic *E. coli* (*Shawki and McCole, 2017*; *Croxen and Finlay, 1999*), but also other *Enterobacteriaceae* (*Struve et al., 2008*; *Kolenda et al., 2019*), use type 1 pili to adhere to host cells in their respective ecological niches (*Spaulding et al., 2017*). Type 1 pili undergo a constant and reversible change in their expression due to phase variation which allows populations of isogenic bacteria to exhibit controlled genotypic and phenotypic variation (*Bayliss, 2009*). In the case of type 1 pili, site-specific recombinases place the *fimA* promoter either in the phase-ON or phase-OFF orientation, resulting in piliated or non-piliated bacteria (*Schilling et al., 2001*). Uropathogenic *E. coli* (UPECs) have mastered the use of phase variation as a remarkable genetic mechanism of plasticity for their advantage, since type 1 pili expression is tightly regulated by the host environment. For example, growing UPECs in vitro in human urine locks expression in the phase-OFF state (*Greene et al., 2015*), whereas adhesion to host cells has been shown to lock expression in the phase-ON state (*Greene et al., 2015*; *Lim et al., 1998*). Phase variation greatly adds to the virulence and fitness of UPECs by generating heterogeneity among the bacterial population where individual cells switch back and forth between the type 1 piliated and non-piliated phenotype (*Bayliss, 2009*; *Wright et al., 2007*). Type 1 pili are necessary for the persistent and therefore recurring infection of the bladder (*Hunstad and Justice, 2010*; *Wright et al., 2007*), but also for other persistent bacterial infections (*Shawki and McCole, 2017*).

Several mechanisms are known to contribute to persistent or recurring infections, one of which is the residing of pathogenic bacteria within a protected niche (*Grant and Hung, 2013*). Such niches can be physical structures when the pathogen invades host cells to hide from the immune response (*Grant and Hung, 2013*; *Donnenberg, 2013*). For example, UPECs invade host epithelium cells in the bladder using type 1 pili, propagate intracellularly and compromise the host defense barriers (*Hunstad and Justice, 2010*; *Martinez et al., 2000*). Strikingly, also the host immune response may unintentionally create protected niches. For example, bladder residing macrophages sequester UPECs before antigen-presenting cells, such as dendritic cells (DCs), come into action (*Mora-Bau et al., 2015*).

DCs are the key cell type connecting the innate and adaptive immune response (*Mellman and Steinman, 2001*). Distinct subtypes of DCs, expressing different surface receptors (*Merad et al., 2013*), reside in every tissue of the host and DCs were also identified at the epithelial junction of the bladder (*Schilling et al., 2003*). In response to an infection, these resident DCs together with newly recruited inflammatory DCs, sense and ingest pathogens. Subsequently, they start to secrete immune-modulatory cytokines and migrate from the site of infection to the draining lymph nodes where they present acquired and processed antigens to T cells, thus triggering the adaptive immune response. However, even if the host immune system manages to detect the pathogen, eradication of the infection is not guaranteed. Pathogens utilize several strategies to subvert innate and adaptive immunity, thereby avoiding clearance by the host and establishing persistent infections (*Donnenberg, 2013*). For example, UPECs were shown to interfere at the interface between innate and adaptive immunity: after contact with UPECs during urinary tract infections (UTIs), tissue resident mast cells secrete high amounts of interleukin-10 (IL-10), an immunosuppressive cytokine (*Chan et al., 2013*), that drives the

differentiation of regulatory T cells (*Hsu et al., 2015*). The combined effect of subverting the host immune response and establishing a protected niche inside the host greatly increases the ability of pathogens to cause persistent infections.

Here, we asked whether type 1 pili play a role in manipulating host immunity and thereby facilitate recurring bacterial infections by dissecting the underlying molecular mechanism between the interaction of UPEC and DCs.

## Results

### Type 1 piliated UPECs inhibit T cell activation and proliferation by decreasing expression of co-stimulatory molecules on DCs

To study the influence of type 1 pili on the adaptive immune system, we genetically engineered bacteria with and without pili. The *fim* genes that produce the molecular components of the type 1 pilus are part of an operon on the *E. coli* chromosome (*Figure 1A, B*) whose expression is regulated by phase variation. We generated stable phase-locked bacterial mutants by locking the fim switch (*fimS*) either in the phase-ON orientation, resulting in constitutive expression of the type 1 pilus, or in the phase-OFF orientation, blocking expression of the pilus (hereafter simply termed ON and OFF, respectively). To achieve this, we deleted the 9-bp long recognition site for the site-specific recombinases FimB and FimE in the internal repeat region upstream of the *fimS* element in either orientation (*Figure 1C*). The presence and absence of type 1 pili was confirmed by electron microscopy and yeast agglutination assay (*Figure 1D*). Additionally, electron microscopy indicated the absence of any other pili, such as p fimbriae, under the chosen growth conditions since the OFF mutant appeared bald (*Figure 1D*).

Since the adaptive immune response seems to be limited during persistent or recurring infections (*Magalhaes et al., 2007*), such as recurring bladder infections (*Abraham and Miao, 2015*), we asked if activation and proliferation of naive T cells in vitro is altered upon stimulating DCs with the genetically constructed UPEC ON or OFF mutants. The activation of ovalbumin (OVA) specific CD4$^+$ T cells (*Banchereau and Steinman, 1998*), as assessed by CD69 receptor upregulation and CD62L receptor downregulation, was massively reduced when DCs were stimulated with OVA and UPEC ON bacteria, compared to OVA and UPEC OFF bacteria (*Figure 1E*). Accordingly, the number of proliferating T cells was also strongly decreased (*Figure 1F*).

DC differentiation and maturation, read out with surface levels of CD11c and MHCII, respectively, were only marginally altered upon exposure to UPEC ON (*Figure 1—figure supplement 1C, D*). Beyond antigen presentation on MHCII, the expression of co-stimulatory molecules on DCs is decisive for effective T cell priming and differentiation into effector cells (*Banchereau and Steinman, 1998*). We therefore analyzed surface expression of CD40, CD80, and CD86 after ON and OFF stimulation and found substantially decreased levels after ON stimulation (*Figure 1G*).

These data suggest that type 1 piliated UPECs prevent effective activation of the adaptive immune response by decreasing surface expression of co-stimulatory molecules on DCs and thus restrict their ability to activate T cells.

### Type 1 piliated UPECs decrease DC migratory capacity and increase T cell contact times by triggering integrin activation

Beyond presentation of antigen together with co-stimulatory factors, two additional cell biological parameters are essential for the priming of T cells. First, DCs have to migrate from the site of antigen encounter and uptake (usually peripheral tissues) to the site of antigen presentation (e.g., lymph nodes), where they meet T cells. This migration step is fully dependent on directional guidance via chemokines, but largely independent of adhesive interactions with the extracellular matrix (*Lämmermann et al., 2008*). Second, T cells have to physically interact with DCs in order to probe their surface for antigen presentation and co-stimulation, meaning that contact dynamics between DCs and T cells are essential parameters determining T cell activation and proliferation (*Bousso, 2008*).

We first investigated DC–T cell contacts using live cell microscopy and found that UPEC ON bacteria, compared to OFF bacteria, increased the antigen specific contact times between DCs and naive CD4$^+$ T cells. Even after 6 hr of co-culture, a large fraction of T cells was unable to dissociate from DCs (*Figure 2A*). This was specific to antigen-bearing DCs, as in the absence of OVA contact

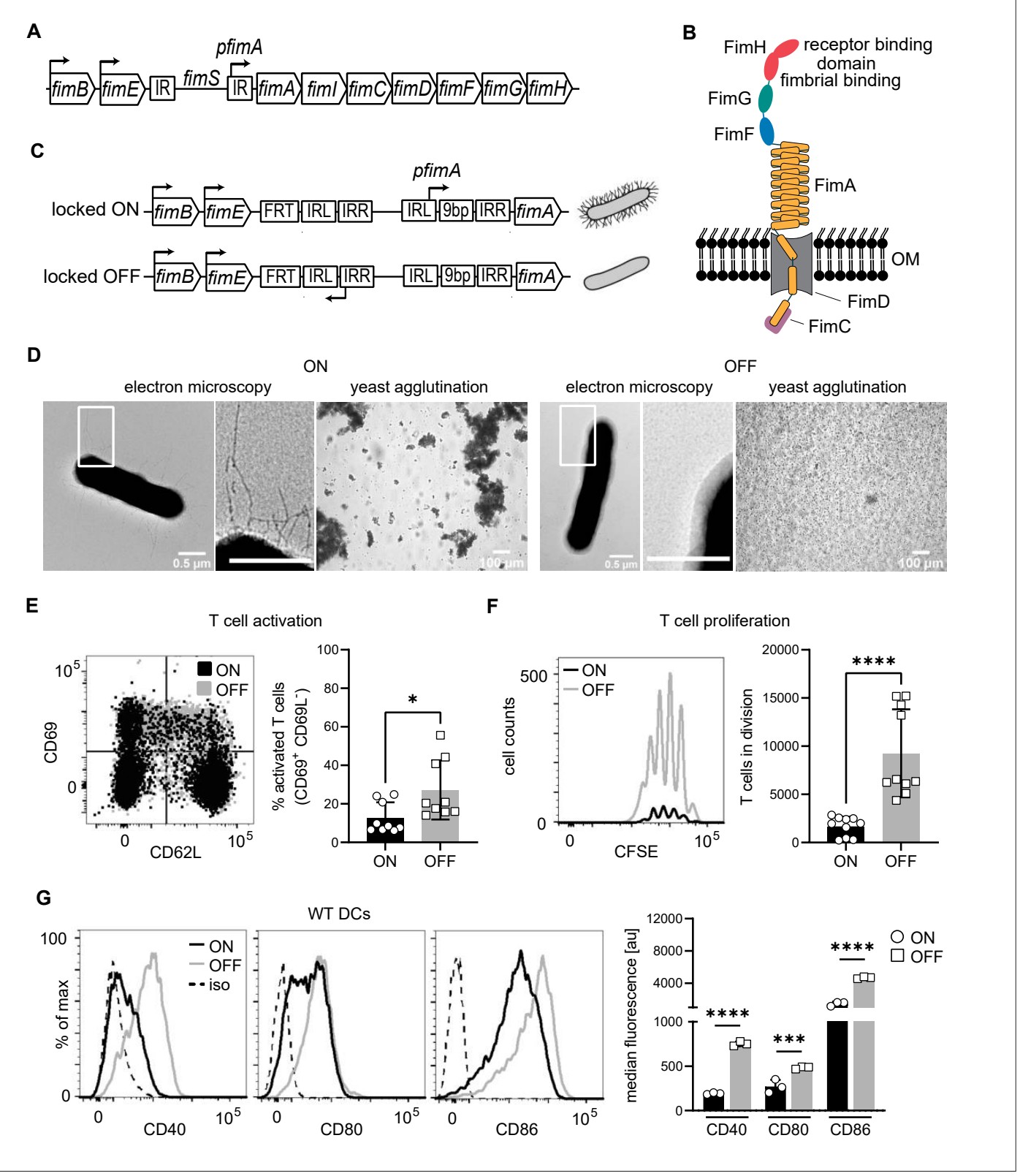

**Figure 1.** Type 1 piliated uropathogenic *E. coli* (UPECs) inhibit T cell activation and proliferation by decreasing co-stimulatory molecules on dendritic cells (DCs). (**A**) Type 1 pili genes are expressed from the *fim* operon. Phase variation of the *fim* switch (*fimS*) harboring *fimA* promoter (*pfimA*) drives expression. *fimB* and *fimE* genes express site-specific recombinases, FimB and FimE, respectively, inverting the *fimS* region by binding to the inverted repeats (IR). (**B**) Type 1 pili consist of several repeating units of the rod protein FimA, two adaptor proteins FimF and FimG, and the tip protein FimH.

*Figure 1 continued on next page*

*Figure 1 continued*

Fimbrial- and a receptor-binding domain of the two-domain FimH protein are shown. FimD, outer membrane usher. FimC, chaperone. (**C**) Phase-locked mutants were generated by deleting the 9-bp recognition site for the site-specific recombinases in the left inverted repeat region of the *fimS* in either ON or OFF orientation by introducing FRT sites resulting in piliated and non-piliated bacteria (see Methods). (**D**) ON (left panel) and OFF mutants (right panel). Electron microscopy images, with zoomed in regions (white boxes) marked in inlays, and yeast agglutination assay. (**E**) Dot plot of CD69 and CD62L expression on T cells after co-culture with ON (black) and OFF (gray) stimulated DCs (left panel). Quantification of CD69+ CD62L− T cell frequencies (right panel) (four biological replicates). (**F**) CFSE (carboxyfluorescein succinimidyl ester) dilution profile of T cells after 96 hr of co-culture with ON (black) and OFF (gray) stimulated wild-type (WT) DCs (left). Quantification of T cells in division (right panel) (four biological replicates). (**G**) Expression level of co-stimulatory molecules (CD40, CD80, and CD86) of WT DCs after stimulation with ON (black) and OFF (gray) (left panel; iso – isotype control). Quantification of median fluorescence values of co-stimulatory molecules (right panel) (three biological replicates). *p < 0.1, ***p < 0.01, ****p < 0.001 by Student's *t*-test; data are represented as means ± standard deviation (SD).

The online version of this article includes the following source data and figure supplement(s) for figure 1:

**Source data 1.** In vitro T cell assay – T cell activation after contact with wild-type (WT) dendritic cells (DCs) after ON and OFF stimulation.

**Source data 2.** In vitro T cell assay – T cells in division after contact with wild-type (WT) dendritic cells (DCs) after ON and OFF stimulation.

**Source data 3.** Co-stimulators on wild-type (WT) dendritic cells (DCs) after ON and OFF stimulation.

**Figure supplement 1.** In vitro generated dendritic cells (DCs) from Hoxb8-progenitor cells resemble iCD103 DCs and differentiation of cells is not different after uropathogenic *E. coli* (UPEC) ON and OFF stimulation.

**Figure supplement 1—source data 1.** Percent mature dendritic cells (DCs) after ON and OFF stimulation.

**Figure supplement 1—source data 2.** MHCII expression after ON and OFF stimulation.

**Figure supplement 2.** Uropathogenic *E. coli* (UPEC) ON and OFF bacteria exhibit slightly different growth rates but no difference in minimal inhibitory concentration (MIC) to gentamicin.

**Figure supplement 2—source data 1.** Growth rate and doubling time of ON and OFF uropathogenic *E. coli* (UPEC).

**Figure supplement 2—source data 2.** Minimal inhibitory concentration (MIC) in R10H20 medium of ON and OFF uropathogenic *E. coli* (UPEC).

times were short and indistinguishable between ON and OFF stimulation. Activation of β2 integrins, specifically CD11b, on DCs was shown to increase the duration of cell–cell contacts by binding to its counter receptor ICAM-1 on T cells, leading to a decrease in the activation of T cells (*Varga et al., 2007*). We therefore tested if the activation status of CD11b on DCs was affected after stimulation with ON bacteria. We analyzed total and active levels of CD11b using the activation-independent antibody M1/70 and CBRM1/5 antibody, which recognizes the active conformation of CD11b (*Oxvig et al., 1999* and *Figure 2—figure supplement 1A*). We found that UPEC ON stimulation shifted CD11b to the active conformation when compared to UPEC OFF or LPS stimulation (*Figure 2B* and *Figure 2—figure supplement 1A*).

In line with the finding that type 1 piliated UPEC enhanced CD11b activity, ON bacteria triggered tight adhesion of DCs to serum-coated surfaces (*Figure 2C*). Similar to the stabilized cell–cell contacts, this surface immobilization was integrin mediated, as for β2 integrin knockout DCs the differential adhesion was lost (*Figure 2C* and *Figure 2—figure supplement 1B*). (Notably, β2 integrin-deficient DCs showed increased β1 integrin-mediated background binding [*Figure 2—figure supplement 1D*].) In line with their excessive surface immobilization, ON stimulated DCs were largely immobile on 2D surfaces and migratory speed was reduced compared to OFF stimulated DCs (*Figure 2—figure supplement 1C*). Next, we performed in vitro migration assays where DCs migrate in 3D cell-derived matrices (*Kaukonen et al., 2017*) and found diminished migration after stimulation with ON bacteria (*Figure 2D*). The same effect on the migratory capacity we observed in ear crawl-out assays where endogenous skin DCs migrate within their physiological tissue in situ (*Stösel et al., 1997*). Here, the ventral halves of explanted mouse ears were repeatedly exposed to either UPEC ON or OFF bacteria during 48 hr. After stimulation with ON bacteria, fewer endogenous DCs were found inside lymph vessels as compared to OFF stimulation (*Figure 2E*). In ears harvested from β2 knockout mice the migration defect was rescued and similar levels of *Itgb2*−/− DCs migrated into the lymph vessels after ON and OFF stimulation (*Figure 2E* and *Figure 2—figure supplement 1E*).

These data suggest that hyperactive CD11b hinders both migration and T cell activation of DCs by immobilizing them to extracellular matrix proteins like fibrinogen or cellular ligands like ICAM-1. This integrin gain of function phenotype is in line with findings where pharmacological approaches which activate integrins have stronger in vivo effects on leukocyte migration than approaches that inhibit integrin function (*Maiguel et al., 2011*). To test if immobilization by hyperactive CD11b is causative

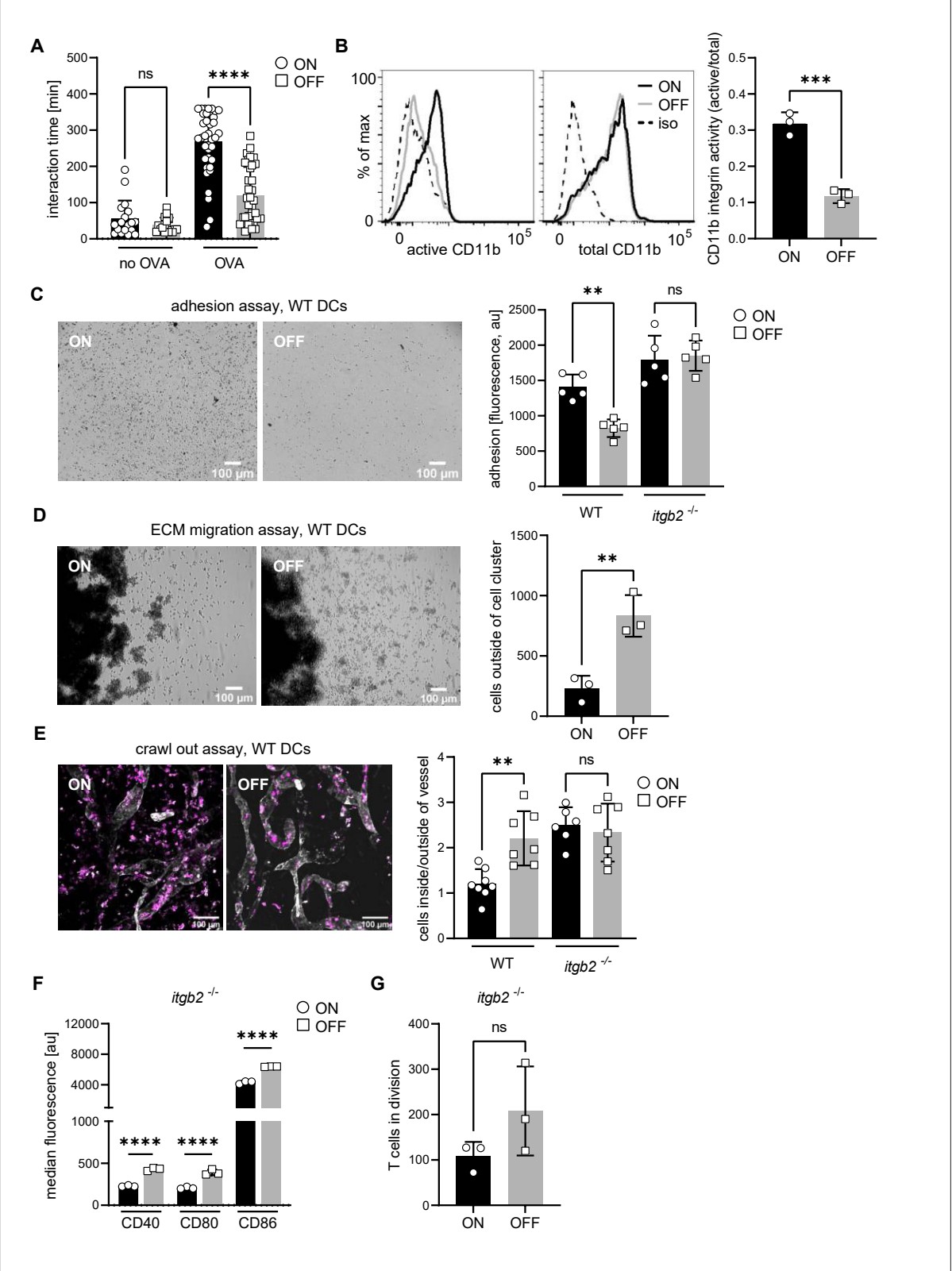

**Figure 2.** Overactivation of integrins increases dendritic cell (DC)–T cell contact time and adhesion of DCs to extracellular matrixes leading to decreased migratory capacity. (**A**) Interaction time between ON and OFF stimulated DCs and T cells in the presence and absence of ovalbumin (OVA) peptide (two biological replicates). (**B**) Histograms of active and total CD11b integrin after ON (black) and OFF (gray) stimulation of DCs (left panel; iso – isotype control). Quantification of CD11b activity (active/total levels of CD11b) (right panel) (three biological replicates). (**C**) Adhesion assay of

*Figure 2 continued on next page*

*Figure 2 continued*

wild-type (WT) DCs after ON and OFF stimulation (left panel). Quantification of fluorescence signals, proxy for adherent cells, after ON (black) and OFF (gray) stimulation of WT and *Itgb2*$^{-/-}$ DCs (right panel) (five biological replicates) (see also **Figure 2—figure supplement 1**). (**D**) Extracellular matrix (ECM) migration assay of WT DCs after ON and OFF stimulation (left panel). Quantification of individual cells outside of cell cluster (right panel) (three biological replicates). (**E**) Ear crawl-out assay of WT DCs after ON and OFF stimulation. Endogenous DCs stained with anti-MHCII (magenta). Lymph vessels stained with anti-LYVE-1 (white) (left panel). Ratios of fluorescent signal inside and outside of lymph vessel after ON (black) and OFF (gray) stimulation of WT and *Itgb2*$^{-/-}$ DCs (right panel) (three biological replicates) (see also **Figure 2—figure supplement 1**). (**F**) Quantification of median fluorescence values of co-stimulatory molecules (CD40, CD80, and CD86) of *Itgb2*$^{-/-}$ DCs after stimulation with ON (black) and OFF (gray) (three biological replicates) (see also **Figure 2—figure supplement 1**). (**G**) Quantification of T cells in division after co-culture with *Itgb2*$^{-/-}$ DCs (three biological replicates) (see also **Figure 2—figure supplement 1**). ns, not significant, **p < 0.05, ***p < 0.01, ****p < 0.001 by one-way analysis of variance (ANOVA) followed by Dunnett's multiple comparisons (A, C, and E) and by Student's *t*-test (B, D, F, and G); data are represented as means ± standard deviation (SD).

The online version of this article includes the following source data and figure supplement(s) for figure 2:

**Source data 1.** Dendritic cell (DC)–T cell contact times after ON and OFF stimulation.

**Source data 2.** CD11b integrin staining of wild-type (WT) dendritic cells (DCs) after ON and OFF stimulation.

**Source data 3.** Adhesion assay of wild-type (WT) dendritic cells (DCs) – fluorescent values after ON and OFF stimulation.

**Source data 4.** Extracellular matrix (ECM) migration assay of wild-type (WT) dendritic cells (DCs) – single cells outside of cell cluster after ON and OFF stimulation.

**Source data 5.** Ear crawl-out assay after ON and OFF stimulation.

**Source data 6.** Co-stimulators on *Itgb2*$^{-/-}$ dendritic cells (DCs) after ON and OFF stimulation.

**Source data 7.** In vitro T cell assay – T cells in division after contact with *Itgb2*$^{-/-}$ dendritic cells (DCs) after ON and OFF stimulation.

**Figure supplement 1.** Type 1 piliated uropathogenic *E. coli* (UPECs) do not affect integrin-independent migration.

**Figure supplement 1—source data 1.** CD11b integrin expression of wild-type (WT) dendritic cells (DCs) after lipopolysaccharide (LPS) and LPS + Mn treatment.

**Figure supplement 1—source data 2.** Migration speed of adherent dendritic cells (DCs) after ON and OFF stimulation.

**Figure supplement 1—source data 3.** Beta1 integrin activity on wild-type (WT) and *Itgb2*$^{-/-}$ dendritic cells (DCs) after ON and OFF stimulation.

**Figure supplement 1—source data 4.** Migration in 3D collagen after ON and OFF stimulation.

for the loss of migration upon UPEC ON stimulation, we measured migration of DCs in 3D collagen gels. Here, DCs efficiently migrate in an integrin-independent manner (**Lämmermann et al., 2008**) and hyperactive CD11b, which does not bind to collagen 1, should no longer be able to immobilize ON stimulated cells. Indeed, we found that the migration speed of ON and OFF stimulated DCs in 3D collagen was indistinguishable (**Figure 2—figure supplement 1F**).

Finally, we asked if activation of β2 integrins maps upstream of the observed detrimental effect on co-stimulatory molecule expression after stimulation with UPEC ON bacteria. *Itgb2*$^{-/-}$ DCs still expressed lowered levels of co-stimulatory molecules after ON stimulation when compared to OFF stimulation (**Figure 2F** and **Figure 2—figure supplement 1G**), suggesting that downregulation of co-stimulatory molecules does not depend on integrin activation. Although *Itgb2*$^{-/-}$ DCs were less efficient in supporting T cell proliferation in vitro (**Figure 2G** and **Figure 2—figure supplement 1H**) when compared to WT DCs (as reported previously in vivo; **Wu et al., 2018**), ON stimulated *Itgb2*$^{-/-}$ DCs were even weaker inducers of T cell proliferation as OFF stimulated *Itgb2*$^{-/-}$ DCs. This suggests that, once DCs encounter T cells, the defect in T cell proliferation is primarily caused by the lack of co-stimulation after UPEC ON stimulation and only to a lesser degree by overactivation of CD11b.

Taken together, type 1 piliated UPEC increase integrin activity on DCs, leading to increased cell–cell and cell–matrix attachment. This causes prolonged interaction times with T cells and impaired migratory capacity. Together, with the above finding of reduced co-stimulatory molecule expression, these data demonstrate that type 1 piliated UPEC target three functional hallmarks of DCs that are critical for the activation of naive T cells: (1) migration to the lymph node, (2) the physical interaction with T cells, and (3) the expression of co-stimulatory molecules.

## The GPI-anchored glycoprotein CD14 binds FimH, making it a novel target for type 1 pili

To find the interaction partner for type 1 pili on host cells, we first investigated the role of the major immune cell receptor TLR4 which was previously suggested as a receptor for the FimH protein of type 1 pili (*Mossman et al., 2008*). We analyzed *Tlr4*$^{-/-}$ DCs after stimulation with UPEC ON bacteria and found that both adhesion and migration followed the pattern of ON stimulated WT cells (*Figure 3A, B* and *Figure 3—figure supplement 1A, B*). Next, we analyzed expression of co-stimulatory molecules on *Tlr4*$^{-/-}$ DCs. It was previously shown that *Tlr4*$^{-/-}$ DCs express reduced levels of co-stimulatory molecules (*Shen et al., 2008*) and this was also true for our in vitro generated *Tlr4*$^{-/-}$ DCs, which hardly expressed any CD40 when compared to the isotype control level (*Figure 3—figure supplement 1C*). Importantly, CD80 and CD86 expression was still reduced after ON stimulation when compared to OFF stimulation (*Figure 3C* and *Figure 3—figure supplement 1C*). Therefore, our results, together with recent findings in *Salmonella* (*Uchiya et al., 2019*), suggest that other receptors besides TLR4 serve as molecular targets of type 1 pili.

We next focused on CD14, a GPI-anchored glycoprotein and co-receptor of TLR4, since a strong correlation has been reported between CD14 expression, integrin activity, and cell adhesion (*Wright et al., 1991*). We performed adhesion and crawl-out migration assays stimulating *Cd14*$^{-/-}$ DCs with UPEC ON and OFF bacteria, and found that both DC functions were fully restored (*Figure 3D, E* and *Figure 3—figure supplement 1D, E*). Additionally, *Cd14*$^{-/-}$ DCs showed almost full rescue of co-stimulatory molecules after ON stimulation, compared to OFF stimulation, with only CD40 still slightly affected (*Figure 3F* and *Figure 3—figure supplement 1F*). Importantly, TLR4-dependent cytokines, such as IL-6, TNF-α, and the interferon-stimulated CCL5 (*Ciesielska et al., 1999*), were not different between ON and OFF stimulated WT DCs (*Figure 3G*) or *Cd14*$^{-/-}$ DCs (*Figure 3—figure supplement 1G*). Thus, the absence of CD14 did not inhibit DC activation by the bacteria and binding of FimH to CD14 did not inhibit the LPS-induced function of CD14 upon exposure to bacteria. Next, we asked if CD14 endocytosis or canonical CD14 downstream targets were altered upon ON stimulation. We could not detect any interferon-alpha 1 expression, regardless of the bacterial stimulus (*Figure 3—figure supplement 1H*) and ON bacteria did not inhibit internalization or synthesis of CD14 when compared to OFF bacteria (*Figure 3H*). Given these results, we assume that binding of FimH to CD14 does not alter the signaling capacity of CD14 to drive interferon-dependent responses upon LPS encounter. Finally, although integrin activity was reduced in *Cd14*$^{-/-}$ DCs (*Figure 3I* and *Figure 3—figure supplement 1I*), UPEC ON bacteria still slightly increased CD11b activity on *Cd14*$^{-/-}$ cells when compared to UPEC OFF bacteria, showing that as opposed to downregulated co-stimulators, integrin activation did not strictly depend on the presence of CD14.

We next asked how type 1 pili interact with the CD14 receptor. In silico protein–protein docking analysis predicted strong binding of −56.98 kcal/mol between FimH (PDB: 6GTV; *Sauer et al., 2019*) and CD14 (PDB: 1WWL; *Kim et al., 2005*, *Supplementary file 1*; *Supplementary file 3*), which is stronger than the −49.81 kcal/mol between FimH and TLR4 (PDB: 3VQ2; *Ohto et al., 2012*) or the −35.55 kcal/mol for CD48 (PDB: 2PTV; *Velikovsky et al., 2007*), another FimH receptor (*McArdel et al., 2016*; *Supplementary file 3*). Binding sites in FimH were located in its N-terminal domain and in CD14 within the central region of the crescent shaped monomer, in an area not overlapping with the LPS-binding region (*Kim et al., 2005*; *Figure 4A* and *Supplementary file 4*).

Since CD14 of mouse (PDB: 1WWL) and human (PDB: 4GLP; *Kelley et al., 2013*) have highly similar secondary structures, but differ in their amino acid sequence (*Kelley et al., 2013*), we performed docking analysis with human CD14. Predicted strong binding of FimH (−55.95 kcal/mol) was conserved in similar regions of both proteins but mediated by different amino acids (*Supplementary file 3* and *Supplementary file 5*). This indicates that the difference in amino acid sequence between mouse and human CD14 is negligible for the otherwise conserved binding between FimH of type 1 pili and CD14.

To verify the predicted binding of FimH to CD14 in vitro, we generated UPEC ON mutants lacking the *fimH* gene (ON Δ*fimH*). Compared to *Salmonella*, where FimH is necessary for the biosynthesis of type 1 fimbriae (*Zeiner et al., 2012*), *E. coli* mutants lacking *fimH* still express type 1 pili which are, however, non-functional (*Maurer and Orndorff, 1985*; *Klemm and Christiansen, 1987*; *Jones et al., 1995*). We confirmed that our ON Δ*fimH* mutants still express type 1 pili but lack the ability to agglutinate with yeast (*Figure 4—figure supplement 1A*). We performed an immunoprecipitation-type approach using magnetic Protein A beads, a CD14-Fc chimeric protein and the bacterial mutants.

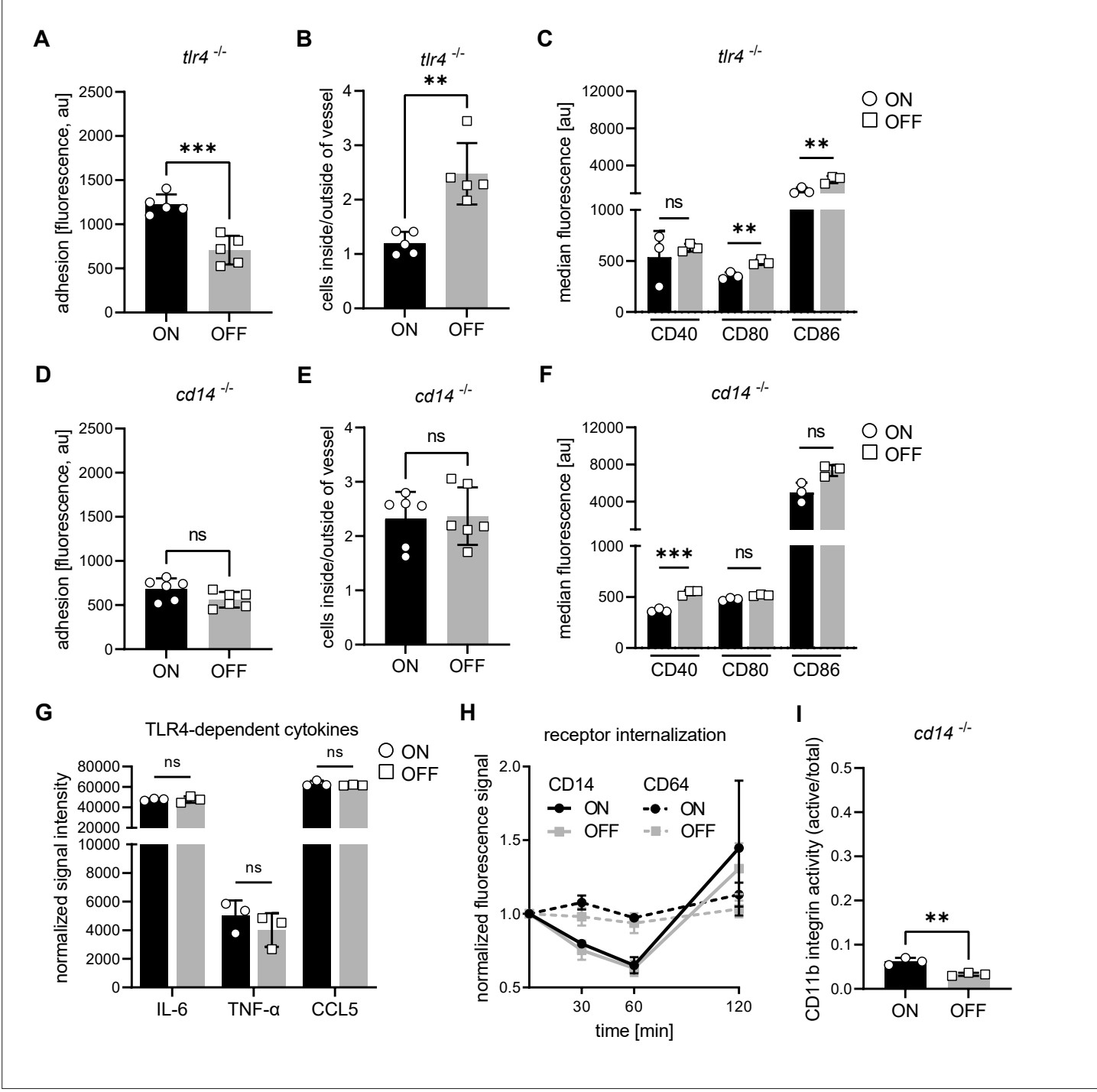

**Figure 3.** Interaction of type 1 piliated uropathogenic *E. coli* (UPEC) with CD14, but not TLR4, is important for the observed phenotypes. (**A**) Adhesion assay of *Tlr4⁻/⁻* dendritic cells (DCs). Quantification of fluorescence signal after ON (black) and OFF (gray) stimulation (five biological replicates) (see also ***Figure 3—figure supplement 1***). (**B**) Ratios of fluorescent signal inside and outside of lymph vessel after ON (black) and OFF (gray) stimulation *Tlr4⁻/⁻* DCs (two biological replicates) (see also ***Figure 3—figure supplement 1***). (**C**) Quantification of median fluorescence values of co-stimulatory molecules (CD40, CD80, and CD86) on *Tlr4⁻/⁻* DCs stimulated with ON (black) and OFF (gray) (three biological replicates) (see also ***Figure 3—figure supplement 1***). (**D**) Quantification of the adhesion assay of *Cd14⁻/⁻* DCs after ON (black) and OFF (gray) stimulation (six biological replicates) (see also ***Figure 3—figure supplement 1***). (**E**) Ratios of fluorescent signal inside and outside of lymph vessel after ON (black) and OFF (gray) stimulation *Cd14⁻/⁻* DCs (three biological replicates) (see also ***Figure 3—figure supplement 1***). (**F**) Quantification of median fluorescence values of co-stimulatory molecules (CD40, CD80, and CD86) on *Cd14⁻/⁻* DCs stimulated with ON (black) and OFF (gray) (three biological replicates) (see also ***Figure 3—figure supplement 1***). (**G**) Cytokine production of ON (black) and OFF (gray) stimulated DCs. Normalized signal intensities of TLR4-dependent cytokines are

*Figure 3 continued on next page*

*Figure 3 continued*

shown (three biological replicates). (**H**) Receptor endocytosis on wild-type (WT) DCs stimulated with ON (black) and OFF (gray). CD14 receptor (solid lines) was stained on DCs fixed at indicated time points. CD64 receptor (dashed lines) served as endocytosis control (three biological replicates). (**I**) Quantification of CD11b activity (active/total levels of CD11b) of *Cd14*$^{-/-}$ DCs stimulated with ON (black) and OFF (gray) (three biological replicates) (see also *Figure 3—figure supplement 1*). ns, not significant, **p < 0.05, ***p < 0.01 by Student's *t*-test; data are represented as means ± standard deviation (SD).

The online version of this article includes the following source data and figure supplement(s) for figure 3:

**Source data 1.** Adhesion assay *Tlr4*$^{-/-}$ dendritic cells (DCs) – fluorescent values after ON and OFF stimulation.

**Source data 2.** Ear crawl-out assay *Tlr4*$^{-/-}$ mice after ON and OFF stimulation.

**Source data 3.** Co-stimulators on *Tlr4*$^{-/-}$ dendritic cells (DCs) after ON and OFF stimulation.

**Source data 4.** Adhesion assay *Cd14*$^{-/-}$ dendritic cells (DCs) – fluorescent values after ON and OFF stimulation.

**Source data 5.** Ear crawl-out assay *Cd14*$^{-/-}$ mice after ON and OFF stimulation.

**Source data 6.** Co-stimulators on *Cd14*$^{-/-}$ dendritic cells (DCs) after ON and OFF stimulation.

**Source data 7.** Cytokine array – TLR4-dependent cytokines after ON and OFF stimulation.

**Source data 8.** Receptor endocytosis – median fluorescence and normalized fluorescence after ON and OFF stimulation.

**Source data 9.** CD11b integrin staining of *Cd14*$^{-/-}$ dendritic cells (DCs) after ON and OFF stimulation.

**Figure supplement 1.** FimH and CD14, but not TLR4, are important for the observed adhesion phenotype.

**Figure supplement 1—source data 1.** Cytokine array – TLR4-dependent cytokines from *Cd14*$^{-/-}$ dendritic cells (DCs) after ON and OFF stimulation.

**Figure supplement 1—source data 2.** Interferon-alpha 1 enzyme-linked immunosorbent assay (ELISA) after ON and OFF stimulation.

First, the CD14-Fc chimera was coupled to the magnetic Protein A beads (*Figure 4—figure supplement 2A*). Next, we introduced a constitutively expressed *yfp* fluorescent marker into the chromosome of the UPEC ON and ON *ΔfimH* mutants for tracking. Using fluorescence microscopy, we found abundant ON bacteria bound per CD14-coupled bead, whereas binding of the *fimH* deletion mutants was scarce (*Figure 4B*). Binding of ON bacteria was not due to unspecific binding of FimH to the bead matrix, as uncoupled beads also showed very scarce binding. Additionally, we analyzed the binding of bacteria to beads by flow cytometry by gating on the size parameters and fluorescence signal of the bacteria population (see *Figure 4—figure supplement 2B* for gating strategy and methods section for further technical details). ON bacteria showed increased binding to CD14-coupled beads when compared to the *fimH* deletion mutants or to ON bacteria binding to uncoupled beads (*Figure 4C*). To further test if only FimH is necessary for binding to CD14, and not interactions with LPS on the bacterial membrane, we extracted type 1 pili (*Sheikh et al., 2017*) from UPEC ON and ON *ΔfimH* mutants and performed a dot blot assay using biotinylated CD14 and streptavidin–HRP (*Figure 4D*). Biotinylated CD14 only bound to type 1 pili extracts from ON bacteria, whereas no binding was observed to type 1 pili extracts from ON mutants lacking the FimH protein.

We next tested if binding of FimH to CD14 is causative for the increase in adhesion of DCs and the decrease in co-stimulator molecules on their membrane after UPEC ON stimulation. Indeed, deleting *fimH* from UPEC ON bacteria fully rescued the adhesion and the expression of all co-stimulatory molecules when compared to stimulation with the UPEC ON bacteria (*Figure 4E, F* and *Figure 4—figure supplement 1B, C*).

Next, we asked whether expression of pathogenic FimH in an otherwise non-pathogenic bacterial background is sufficient to affect expression of co-stimulatory molecules on DCs. We constructed a locked-ON mutant of *E. coli* W, a non-pathogenic *E. coli* strain, and replaced the endogenous *fimH* gene with the pathogenic *fimH* variant of the UPEC strain CFT073. This non-pathogenic ON *fimH*$_{CFT073}$ bacteria was able to agglutinate with yeast, confirming the correct insertion of the pathogenic *fimH* gene (*Figure 5—figure supplement 1A*). However, neither adhesion nor expression of co-stimulatory molecules of DCs was affected after stimulation with the non-pathogenic ON *fimH*$_{CFT073}$ mutants as compared to the non-pathogenic *E. coli* W ON bacteria (*Figure 5—figure supplement 1B, C*). Given these results we asked if the observed immune-modulatory properties are specific to the pathogenic strain CFT073 only, or if other UPEC strains similarly affect DC activation. Thus, we constructed ON and OFF mutants of two additional clinical isolates, UTI89 (*Chen et al., 2006*) and 536 (*Hochhut et al., 2005*), and analyzed adhesion and expression of co-stimulatory molecules of DCs. Interestingly, only the ON bacteria of UTI89 strongly increased adhesion of DCs, whereas ON bacteria of 536 had only

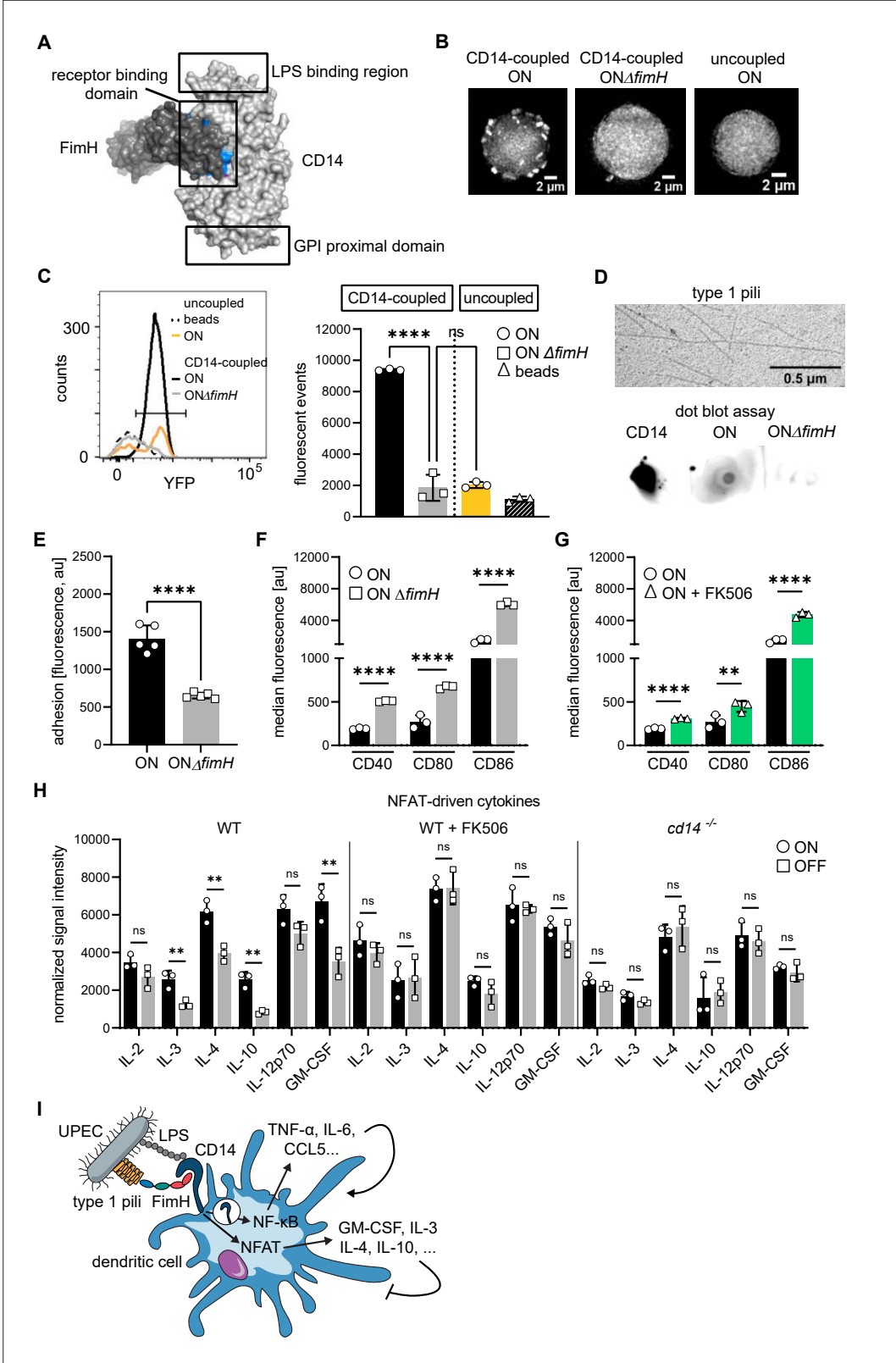

**Figure 4.** FimH binds to CD14 via protein–protein interactions and deletion of *fimH* rescues adhesion and expression of co-stimulatory molecules. (**A**) In silico protein–protein docking analysis for FimH and CD14 (see also ***Supplementary files 1 and 3***). FimH is shown in dark gray and CD14 in light gray. Surface plot of docked proteins from the view of CD14 on the membrane of the cell is shown. GPI proximal domain, lipopolysaccharide

*Figure 4 continued on next page*

*Figure 4 continued*

(LPS)- and receptor-binding domains are indicated. Top 10 amino acids predicted to interact during the binding are highlighted in blue for FimH and magenta for CD14 (most amino acids are buried inside the surface plot). (**B**) Microscopy images of bead-binding assay. ON or ON Δ*fimH* mutants expressing a constitutive *yfp* fluorescent marker bound to CD14-coupled or uncoupled beads are shown. (**C**) Flow cytometry analysis of bead-binding assay. Histogram of fluorescence events in the bacteria gate (left panel). Uncoupled beads (dashed black), ON bacteria bound to uncoupled beads (yellow), ON bacteria bound to CD14-coupled beads (black), ON Δ*fimH* mutants bound to CD14-coupled beads (gray). Quantification of fluorescent events in the bacterial gate (right panel) (three biological replicates) (gating strategy see *Figure 4—figure supplement 2*). (**D**) Type 1 pili extracts and dot blot assay. Electron microscopy images of type 1 pili extracts from ON bacteria (upper panel). Dot blot assay of type 1 pili extracts with biotinylated CD14 (lower panel). Pre-blotted biotinylated CD14 served as positive control. Bound CD14 was visualized with streptavidin–HRP antibody. (**E**) Quantification of fluorescence signals, proxy for adherent dendritic cells (DCs), after ON (black) and ON Δ*fimH* (gray) stimulation (five biological replicates) (see also *Figure 4—figure supplement 1*). ON data are the same as in *Figure 2C*. (**F**) Quantification of median fluorescence values of co-stimulatory molecules (CD40, CD80, and CD86) on wild-type (WT) DCs stimulated with ON (black) and ON Δ*fimH* mutants (gray) (three biological replicates) (see also *Figure 4—figure supplement 1*). ON data are the same as in *Figure 1G*. (**G**) Quantification of median fluorescence values of co-stimulatory molecules (CD40, CD80, and CD86) on WT DCs stimulated with ON (black) or ON after pre-treatment of DCs with 50 mM FK506 for 20 min prior to infection (green) (see also *Figure 4—figure supplement 1*). ON data are the same as in *Figure 1G*. (**H**) Cytokine production of ON (black) and OFF (gray) stimulated WT DCs, WT DCs pretreated with FK506 and *Cd14⁻/⁻* DCs. Normalized signal intensities of NFAT-dependent cytokines are shown (three biological replicates). (**I**) Schematic of uropathogenic *E. coli* (UPEC) binding to DCs. Type 1 piliated UPEC bind to CD14 receptor via FimH and LPS. LPS binding stimulates CD14 endocytosis and NF-$\kappa$B-dependent cytokine expression (TNF-α, IL-6, CCL5, etc.) stimulating DC maturation. FimH binding stimulates NFAT pathway and expression of immune-modulatory cytokines (GM-CSF, IL-3, IL-4, IL-10, etc.). ns, not significant, **p < 0.05, ****p < 0.001 by one-way analysis of variance (ANOVA) followed by Dunnett's multiple comparisons (C) and by Student's *t*-test (E–H); data are represented as means ± standard deviation (SD).

The online version of this article includes the following source data and figure supplement(s) for figure 4:

**Source data 1.** Bead-binding assay – fluorescent events inside bacteria gate after incubation with ON and ON Δ*fimH* bacteria.

**Source data 2.** Adhesion assay of wild-type (WT) dendritic cells (DCs) – fluorescent values after ON and ON Δ*fimH* stimulation.

**Source data 3.** Co-stimulators on wild-type (WT) dendritic cells (DCs) after ON and ON Δ*fimH* stimulation.

**Source data 4.** Co-stimulators on wild-type (WT) dendritic cells (DCs) after ON stimulation with or without FK506.

**Source data 5.** Cytokine array – NFAT-dependent cytokines after ON and OFF stimulation.

**Figure supplement 1.** FimH is necessary for the observed phenotypes and binding of FimH to CD14 triggers overactivation of the NFAT pathway.

**Figure supplement 2.** Bead assay using CD14-coupled magnetic beads.

a moderate effect (*Figure 5—figure supplement 1D*). Additionally, none of the two isolates affected all of the co-stimulatory molecules, but both decreased expression of CD86 only (*Figure 5—figure supplement 1E, F*). Thus, we hypothesize that compared to these two strains, the clinical isolate CFT073 expresses additional virulence factors, which act downstream of the FimH–CD14 interaction to modulate the functional hallmarks of the immune response of DCs.

Finally, we were interested in the molecular mechanism(s) triggered upon FimH binding to CD14. Since NFAT (nuclear factor of activated T cells) signaling was shown to map downstream of CD14-mediated Ca²⁺ influx (*Zanoni et al., 2009*), we used the inhibitor FK506 to block calcineurin prior to stimulation with bacteria. Blocking the NFAT pathway partially rescued expression of co-stimulatory molecules after ON stimulation (*Figure 4G* and *Figure 4—figure supplement 1D*), establishing NFAT as one downstream target of FimH. Thus, we tested if overactivation of the NFAT pathway after UPEC ON stimulation affects production of cytokines known to be NFAT dependent (*Fric et al., 2012*). Indeed, we found several NFAT-target cytokines to be increased after ON compared to OFF stimulation, with IL-3, IL-4, IL-10, and GM-CSF being most severely affected (*Figure 4H*), whereas blocking of the NFAT pathway or deletion of CD14 resulted in equal expression of NFAT-target cytokines between ON and OFF stimulation (*Figure 4H*). However, it should be noted, that blocking of calcineurin with FK506 resulted in a general upregulation of cytokines, most likely because several

NFAT-target cytokines are also regulated by other means such as NF-κB, AP-1, or STAT signaling pathways (*Hermann-Kleiter and Baier, 2005*).

Based on these experimental observations, we propose the following model of interaction between FimH on UPECs and CD14 on DCs (*Figure 4I*): CD14 directly binds FimH on the tip of type 1 pili, leading to overactivation of the NFAT pathway and increased expression of immunomodulatory cytokines, such as GM-CSF, IL-3, IL-4, and IL-10. LPS-induced activation of CD14 is not affected by FimH binding, leaving CD14 endocytosis and expression of proinflammatory cytokines, such as TNF-α, IL-6, and CCL5, unaltered.

## FimH amino acids predicted to bind are highly conserved and are partially located in the mannose-binding domain

The two-domain FimH protein consists of a receptor-binding domain and a pili-binding domain (*Choudhury et al., 1999*). The receptor-binding domain does not only interact with host receptors, such as Uroplakin 1a on bladder epithelial cells, but also highly specifically binds D-mannose which due to its specific interaction has been used in the treatment of UTIs (*Wiles et al., 2008*). The amino acids responsible for binding mannose are located in the mannose-binding pocket (P1, N46, D47, D54, Q133, N135, D140, and F142; *Supplementary file 4*; *Hung et al., 2002*) and the tyrosine gate (Y48, I52, and Y137; *Supplementary file 4*; *Touaibia et al., 2017*). These residues are highly conserved among pathogenic and non-pathogenic *E. coli* strains (*Figure 5A*), whereas other amino acids that affect the flexibility of the FimH protein and therefore facilitate host colonization, were found to be mutated in pathogenic *E. coli* (*Chen et al., 2009*; *Sokurenko et al., 1998*; *Kalas et al., 2017*). Interestingly, the most important amino acids of FimH we predicted to be responsible for binding to CD14 are highly conserved among several different *E. coli* strains, irrespective of their pathogenicity (*Figure 5A*).

To verify the significance of these residues for binding to CD14, we introduced mutations into the three most important amino acids. We exchanged amino acids R98 (binding energy −7.23 kcal/mol), T99 (binding energy −4.92 kcal/mol), and Y48 (binding energy −4.29 kcal/mol) individually to alanine or all three at the same time creating a triple mutant. All four FimH mutants were still able to bind to CD14-coupled beads and only the triple mutant showed a more pronounced decrease in binding (*Figure 5B* and *Figure 5—figure supplement 2A*). Stimulating DCs with the FimH amino acid mutants showed partial rescue for co-stimulatory molecule expression levels (*Figure 5C* and *Figure 5—figure supplement 2B*). This suggests that not only these individual amino acids mediate binding to CD14, but most likely several other amino acids are involved, and thus the supporting secondary structure of FimH as a whole is important for the interaction.

Since Y48 and other identified FimH amino acids with weaker binding energies for CD14 (*Supplementary file 4*) are located in the mannose-binding pocket, we were interested whether FimH antagonists, such as D-mannose or the low molecular weight mannose derivate M4284 (*Schönemann et al., 2019*), disrupt the interaction. Additionally, we tested a blocking CD14 antibody (M14-23) (*Tsukamoto et al., 2010*) for its ability to inhibit binding of FimH to CD14. Our experiments showed that 175 μM D-mannose was not sufficient to block binding of UPEC ON bacteria to CD14, whereas 1 mM D-mannose and M4284 at 10 μM were able to inhibit binding of ON bacteria to CD14-coupled beads (*Figure 6A* and *Figure 6—figure supplement 1A*). The blocking CD14 antibody reduced bacteria binding to beads by roughly 25% but was the most effective at rescuing expression of co-stimulatory molecules on DCs among all tested components (*Figure 6B* and *Figure 6—figure supplement 1B*). Thus, existing FimH antagonists as well as blocking CD14 antibodies interfere with the immunomodulatory action of type 1 piliated pathogens and might be candidates for treating recurring or persistent infections.

Finally, since the CD14 blocking antibody most efficiently rescued expression of co-stimulatory molecules (*Figure 6B*), we assayed its effect on T cell priming in vitro. Proliferation of T cells was strongly enhanced when DCs were treated with 20 μg/ml M14-23, while stimulated with UPEC ON bacteria as compared to ON stimulation only (*Figure 6C*). This confirms the importance of CD14 in the interaction of type 1 piliated UPEC with DCs, and clearly links the interaction of FimH with CD14 to the ability of DCs to mount a functional adaptive immune response.

## Immunosuppression of DCs by type 1 piliated UPEC is not cell autonomous

To test the importance of our in vitro findings in more physiological settings, we devised a series of in vivo experiments. We first performed in vivo T cell priming assays in mice by intravenous injection

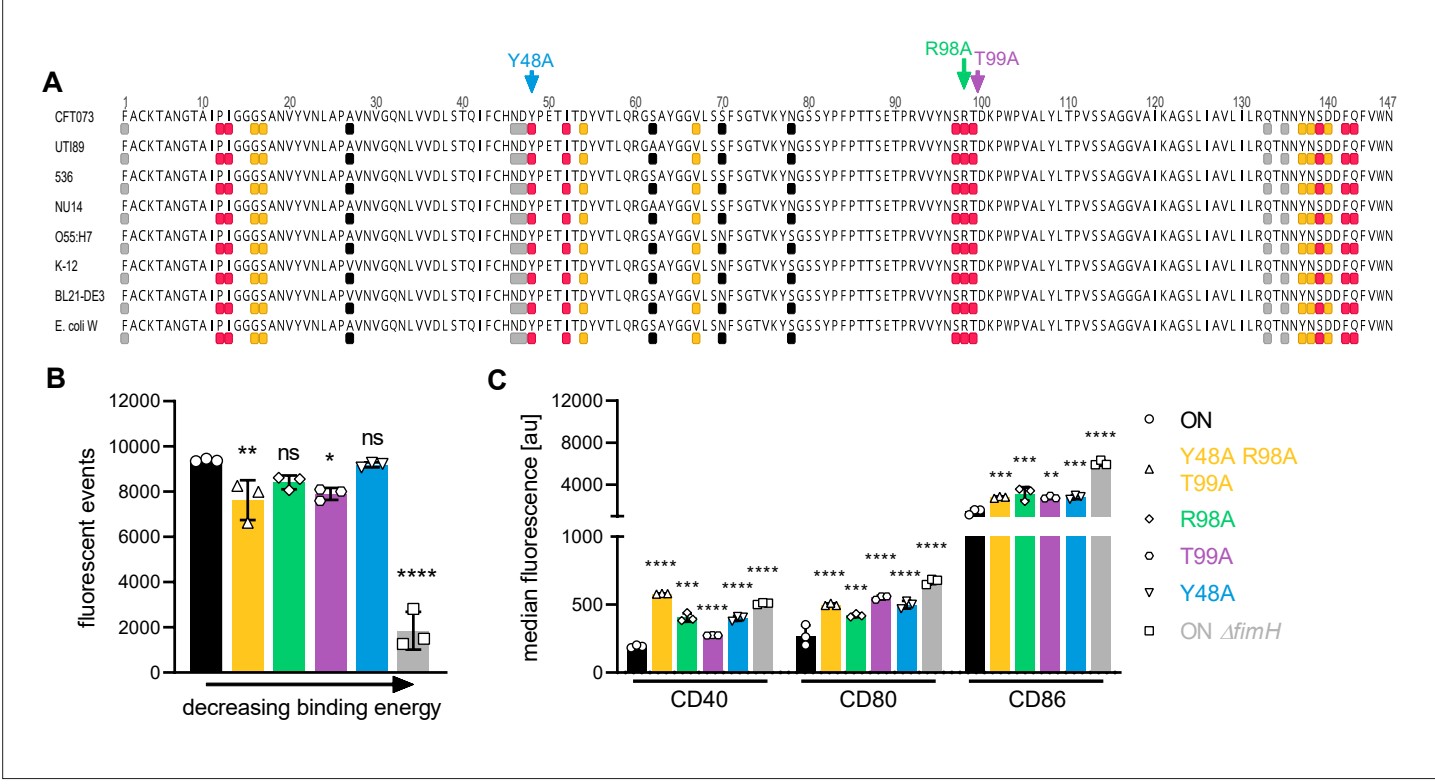

**Figure 5.** FimH binds to CD14 via highly conserved amino acid residues partially located in the mannose-binding domain. (**A**) Receptor-binding domain of FimH from different *E. coli* strains is shown (uropathogenic *E. coli* [UPEC]: CFT073, UTI89, 536, NU14, EPEC: O55:H7; non-pathogenic; K-12, BL21-DE3, *E. coli* W). The top 10 amino acids on FimH showing strongest binding energy toward mouse CD14 (PDB: 1WWL) are shown in pink and toward human CD14 (PDB: 4GLP) are shown in orange. Amino acids I13, P12, and F42 are involved in both, mouse and human CD14, and therefore only shown in pink. Amino acids located in mannose-binding pocket and tyrosine gate are shown in gray. Amino acids I13, Y48, I52, Y137, and F142 are involved in mannose and CD14 binding and therefore only shown in pink. Common pathoadaptive mutations that differ between UPEC and non-pathogenic *E. coli* are shown in black. Amino acids mutated to generate FimH amino acid mutants were Y48 (binding energy −4.29 kcal/mol), T99 (binding energy −4.92 kcal/mol), and R98 (binding energy −7.23 kcal/mol) (see also **Supplementary files 3 and 5**). (**B**) Bead-binding assay of FimH amino acid mutants. Quantification of fluorescent events in the bacterial gate of ON (black), ON *ΔfimH* mutants (gray), and FimH amino acid mutants Y48A (blue), T99A (violet), R98A (green), and Y48A R98A T99A (yellow) (three biological replicates) (see also **Figure 5—figure supplement 2**). ON and ON *ΔfimH* data are the same as in **Figure 4C**. (**C**) Quantification of median fluorescence values of co-stimulatory molecules (CD40, CD80, and CD86) of wild-type (WT) dendritic cells (DCs) after stimulation with ON (black), ON *ΔfimH* mutants (gray), and FimH amino acid mutants Y48A (blue), T99A (violet), R98A (green), and Y48A R98A T99A (yellow) (three biological replicates) (see also **Figure 5—figure supplement 2**). ON and ON *ΔfimH* data are the same as in **Figure 4F**. ns, not significant, *p < 0.1, **p < 0.05, ***p < 0.01, ****p < 0.001 by one-way analysis of variance (ANOVA) followed by Dunnett's multiple comparisons (the mean of the data were compared to the mean of ON); data are represented as means ± standard deviation (SD).

The online version of this article includes the following source data and figure supplement(s) for figure 5:

**Source data 1.** Bead-binding assay – fluorescent events inside bacteria gate after incubation with amino acid mutants in FimH.

**Source data 2.** Co-stimulators on wild-type (WT) dendritic cells (DCs) after stimulation with amino acids mutants in FimH.

**Figure supplement 1.** The quantitative effect depends on uropathogenic *E. coli* (UPEC) strain CFT073, but not on FimH from CFT073 alone.

**Figure supplement 1—source data 1.** Adhesion assay of wild-type (WT) dendritic cells (DCs) – fluorescent values after ON and ON *fimH*$_{CFT073}$ stimulation.

**Figure supplement 1—source data 2.** Expression of co-stimulators on wild-type (WT) dendritic cells (DCs) after ON and ON *fimH*$_{CFT073}$ stimulation.

**Figure supplement 1—source data 3.** Adhesion assay of wild-type (WT) dendritic cells (DCs) – fluorescent values after stimulation with ON and OFF mutants from uropathogenic *E. coli* (UPEC) strains UTI89 and 536.

**Figure supplement 1—source data 4.** Expression of co-stimulators on wild-type (WT) dendritic cells (DCs) after stimulation with ON and OFF mutants from uropathogenic *E. coli* (UPEC) strain UTI89.

**Figure supplement 1—source data 5.** Expression of co-stimulators on wild-type (WT) dendritic cells (DCs) after stimulation with ON and OFF mutants from uropathogenic *E. coli* (UPEC) strain 536.

**Figure supplement 2.** Analysis of amino acid mutants of FimH.

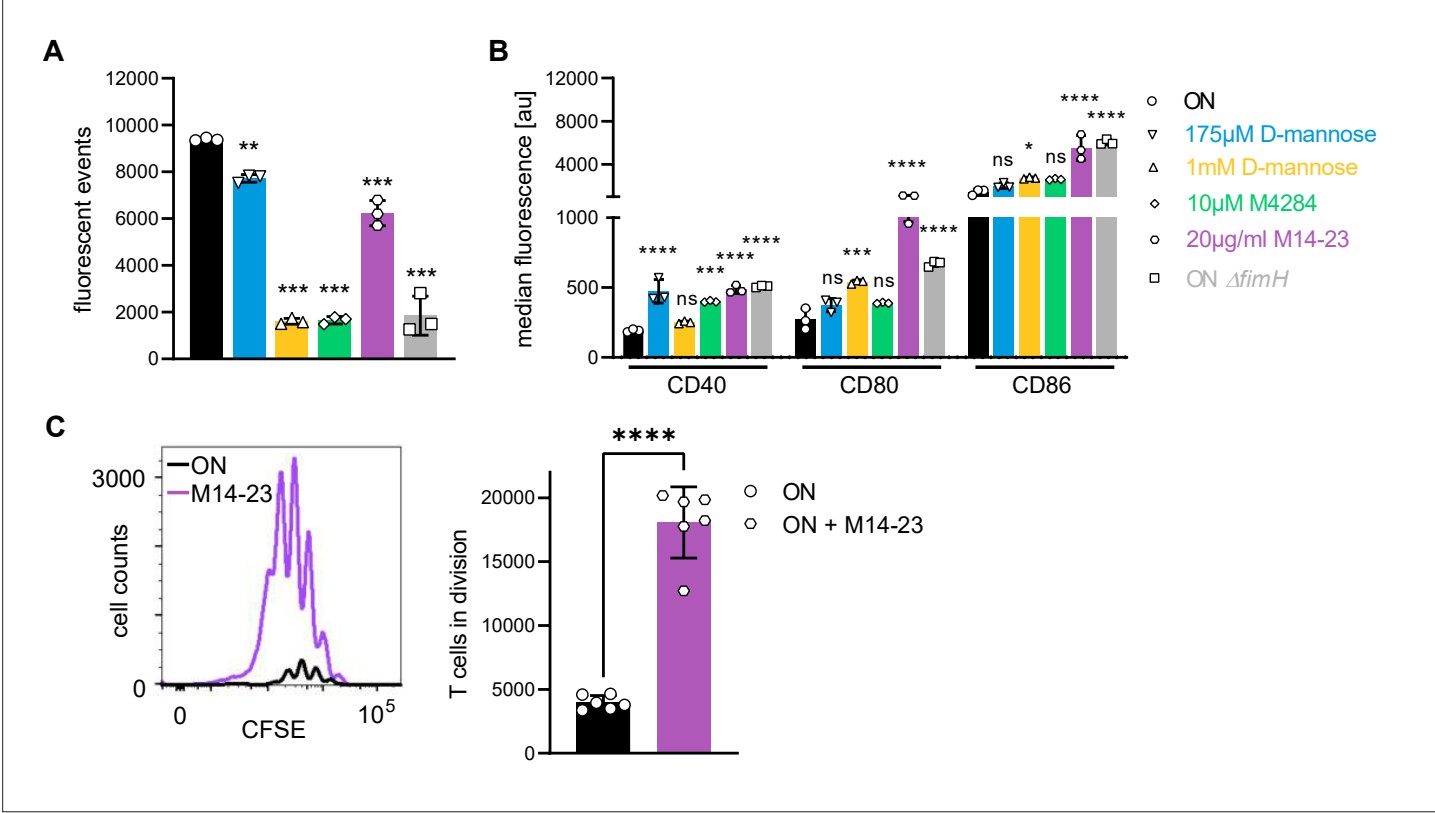

**Figure 6.** FimH antagonists and a blocking CD14 antibody (partially) block binding and rescue expression of co-stimulatory molecules on dendritic cells (DCs). (**A**) Bead-binding assay of ON bacteria in the presence of FimH antagonists and blocking CD14 antibody. Quantification of fluorescent events in the bacterial gate of ON (black), ON *ΔfimH* mutants (gray), 175 µM D-mannose (blue), 1 mM D-mannose (yellow), 10 µM M4284 (green), and 20 µg/ml M14-23 antibody (violet) (three biological replicates) (see also *Figure 6—figure supplement 1*). ON and ON *ΔfimH* data are the same as in *Figure 4E*. (**B**) Quantification of median fluorescence values of co-stimulatory molecules (CD40, CD80, and CD86) of wild-type (WT) DCs after stimulation with ON (black), ON *ΔfimH* mutants (gray), and ON stimulation in the presence of 175 µM D-mannose (blue), 1 mM D-mannose (yellow), 10 µM M4284 (green), and 20 µg/ml M14-23 antibody (violet) (three biological replicates) (see also *Figure 6—figure supplement 1*). ON and ON *ΔfimH* data are the same as in *Figure 4H*. (**C**) CFSE dilution profile of T cells after 96 hr of co-culture with ON (black) and ON in the presence of 20 µg/ml M14-23 antibody (violet) stimulated WT DCs (left panel). Quantification of T cells in division (right panel) (three biological replicates). ns, not significant, *p < 0.1, **p < 0.05, ***p < 0.01, ****p < 0.001 by one-way analysis of variance (ANOVA) followed by Dunnett's multiple comparisons (A, B); the mean of the data were compared to the mean of ON and by Student's *t*-test (C); data are represented as means ± standard deviation (SD).

The online version of this article includes the following source data and figure supplement(s) for figure 6:

**Source data 1.** Bead-binding assay – fluorescent events inside bacteria gate after incubation of ON mutants in the presence of inhibitors.

**Source data 2.** Co-stimulators on wild-type (WT) dendritic cells (DCs) after stimulation with ON mutants in the presence of inhibitors.

**Source data 3.** In vitro T cells in proliferation after contact with wild-type (WT) dendritic cells (DCs) after stimulation with ON mutants and treatment with M14-23.

**Figure supplement 1.** Analysis of FimH antagonists and a blocking CD14 antibody.

of CFSE labeled, OVA peptide-specific CD4 T cells, followed by footpad injections of either ON or OFF stimulated, OVA peptide loaded DCs and analyzed T cell proliferation by flow cytometry. We observed that the same fraction of T cells entered cell division when compared between ON and OFF stimulation. However, priming of T cells by ON stimulated DCs led to significantly fewer T cells in late divisions resulting in a reduced proliferation index, reminiscent of our in vitro findings (*Figure 7A*).

Next, we sought to determine whether the observed defect in T cell priming is due to a reduction in the numbers of ON stimulated DCs that arrive in the lymph node, the intrinsic T cell priming deficiency of ON stimulated DCs, or a combination of both. To this end, we first aimed to resolve the initial events of DC migration, interstitial migration and entry into the lymphatics by devising crawl-in assays where exogenously stimulated DCs are allowed to invade the lymph vessels of the ventral halves of untreated, explanted mouse ears. In line with our previous results, we found fewer ON compared to

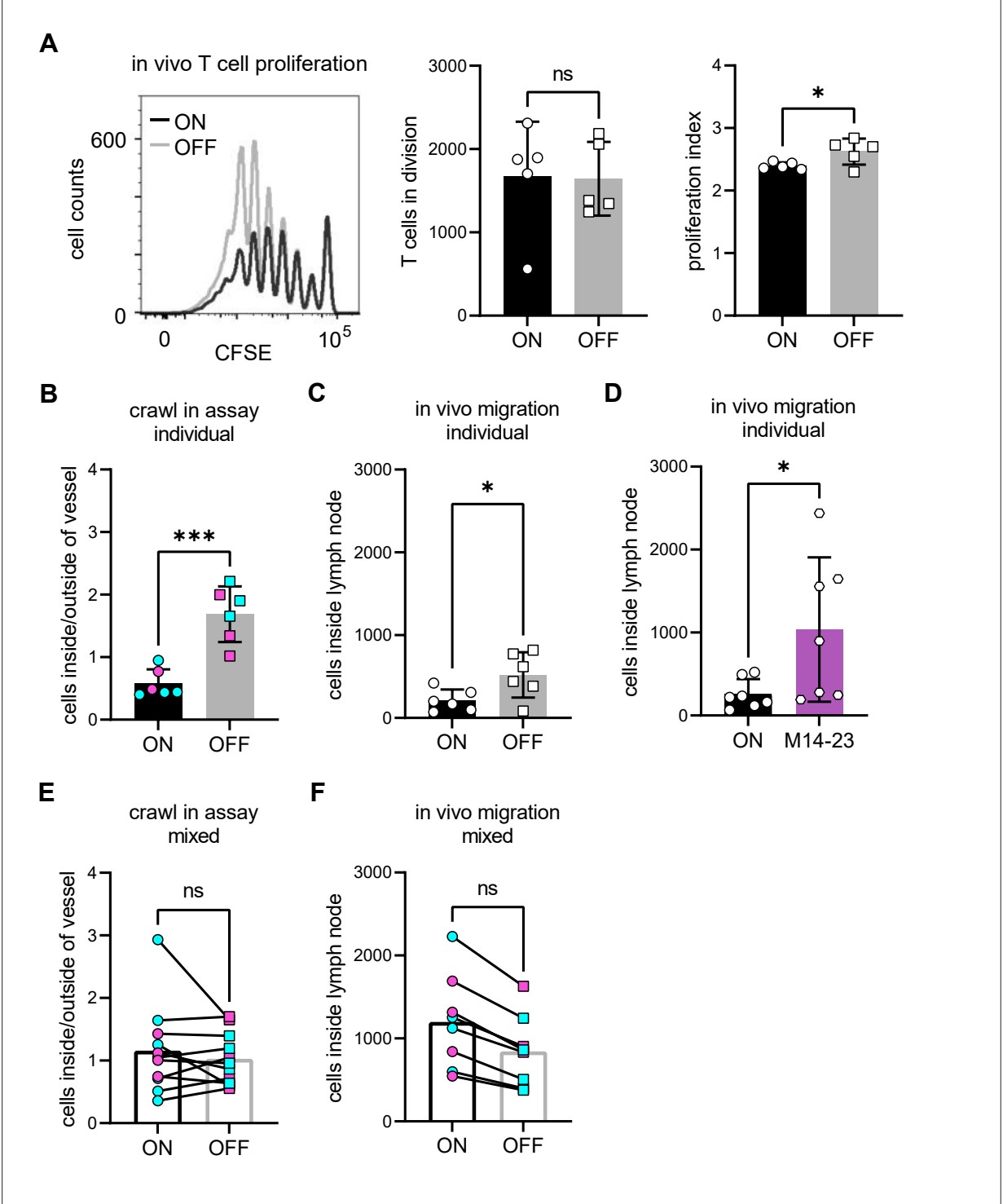

**Figure 7.** Immunosuppression of dendritic cells (DCs) by type 1 piliated uropathogenic *E. coli* (UPEC) is not cell autonomous. (**A**) In vivo CFSE dilution profile of T cells after injection of ON (black) and OFF (gray) stimulated wild-type (WT) DCs (left panel). Quantification of T cells in division and the proliferation index (right panel) (three biological replicates). (**B**) Individual crawl-in assay of ON and OFF stimulated WT DCs. Ratios of fluorescent signal inside and outside lymph vessel after ON (black) and OFF (gray) stimulation. Exogenous DCs were stained with Oregon green (cyan) or TAMRA (Tetramethylrhodamine, magenta) and applied individually onto the ear sheets (six biological replicates). (**C**) Individual in vivo migration of WT DCs. Amount of DCs inside the popliteal lymph node after ON (black) and OFF (gray) stimulation. Exogenous DCs were stained with Oregon green and injected individually into the footpad of WT mice (six biological replicates). (**D**) Individual in vivo migration of WT DCs after treatment with blocking CD14 antibody. Amount of DCs inside the popliteal lymph node after ON stimulation (black) and ON stimulation in the presence of 20 µg/ml M14-23

*Figure 7 continued on next page*

*Figure 7 continued*

antibody (violet). Exogenous DCs were stained with Oregon green and injected individually into the footpad of WT mice (seven biological replicates). (**E**) Mixed crawl-in assays. ON (black) and OFF (gray) stimulated cells were stained with either Oregon green (cyan) or TAMRA (magenta) and mixed in a 1:1 ratio before applying the cells onto the ear sheets (six biological replicates). (**F**) Mixed in vivo migration assays. ON (black) and OFF (gray) stimulated cells were stained with either Oregon green (cyan) or TAMRA (magenta) and mixed in a 1:1 ratio before injected into the footpad of WT mice (four biological replicates). ns, not significant, *p < 0.1, ***p < 0.01 by Student's *t*-test; data are represented as means ± standard deviation (SD).

The online version of this article includes the following source data for figure 7:

**Source data 1.** In vivo T cells in proliferation after contact with wild-type (WT) dendritic cells (DCs) after ON and OFF stimulation.

**Source data 2.** Individual ear crawl in assay – wild-type (WT) mice after ON and OFF stimulation.

**Source data 3.** In vivo individual dendritic cell (DC) migration – cells inside the lymphnode after ON and OFF stimulation.

**Source data 4.** In vivo individual dendritic cell (DC) migration – cells inside the lymphnode after stimulation with ON mutants and treatment with M14-23.

**Source data 5.** Mixed ear crawl in assay – wild-type (WT) mice after ON and OFF stimulation.

**Source data 6.** Mixed in vivo migration assay – cells inside the lymphnode normalized to preinjection mix after ON and OFF stimulation.

OFF stimulated DCs inside the lymph vessels (*Figure 7B*). Next, we aimed to resolve the later stage events of DC migration by in vivo migration assays, in which fluorescently labeled ON and OFF stimulated DCs were injected separately into hind footpads of mice and the total number of cells that arrived in the popliteal lymph node after 72 hr were assessed. Interestingly, we observed only slight reduction in the number of ON stimulated DCs in the lymph nodes compared to OFF stimulated ones (*Figure 7C*). Notably, antibody-mediated blockage of CD14 fully restored migration of ON stimulated DCs in vivo (*Figure 7D*). The alleviated effect at later time points, when DCs were exposed to inflamed tissue, indicated that non-cell autonomous factors can modulate the effect of piliated bacteria. We therefore mixed ON vs. OFF stimulated DCs, stained with different colors, and then either performed crawl-in assays or footpad injections (*Figure 7E, F*). Strikingly, we could no longer observe any differences in the migration of ON and OFF stimulated DCs. This result points toward a strong, non-cell autonomous effect where differentially stimulated DCs mutually affect their migratory capabilities and thus average the binary response observed at the single-cell level, most likely due to exchange of immunomodulatory cytokines (*Figure 4H*). Interestingly, similar non-cell autonomous behavior of DCs integrating information at the population rather than the single-cell level was recently reported in the context of antiviral responses. Here, a quorum of sensed stimuli governed the activation of DCs via secreted cytokines (*Bardou et al., 2021*).

## Discussion

Here, we uncovered that type 1 piliated UPECs target the glycoprotein CD14 on the surface of DCs and thereby shut down the migratory capacity of DCs and their ability to interact with and activate T cells.

It has been proposed that the unusually tight regulation of UPECs type 1 pili expression has evolved to limit exposure to the host immune system, allowing the pathogen to establish a persistent infection (*Donnenberg, 2013*). In this study, we uncovered and characterized a different fundamental role of type 1 pili, as modulators of the innate and adaptive immune response. We found that type 1 piliated UPECs decreased the migratory capacity of DCs by increasing their adhesion to other cells, such as T cells, and the extracellular matrix by overactivation of integrins. The effective and timely transition from an adhesion to a migration phenotype is essential for DCs to migrate from the site of the actual infection to the lymph node in order to interact with lymphocytes. To achieve these migratory and signaling tasks, DCs need to dynamically regulate integrin-mediated adhesion, and we found that overactivation of integrins triggered by type 1 piliated UPECs leads to effective immobilization to the extracellular matrix and decreased turnover of cell–cell interactions. Overactivation of integrins, by hijacking integrin-linked kinases leading to decreased turnover of focal adhesions, was already shown to subvert innate immunity during *Shigella* and enterohemorrhagic *E. coli* (EHEC) infections (*Kim et al., 2009*; *Shames et al., 2010*), however UPECs do not express the same genes. Moreover, we found that type 1 piliated UPECs also decreased the expression of co-stimulatory molecules, which are essential for T cell activation and proliferation.

We identified CD14, a GPI-anchored glycoprotein, as the direct target for the FimH protein of type 1 pili. In analogy, a previous study showed that the *fimA*-encoded major type V fimbriae of the oral pathogen *Porphyromonas gingivalis* also bind CD14 and thereby increase integrin-mediated adhesion by activating CD11b integrin (*Harokopakis and Hajishengallis, 2005*). Using in silico protein–protein docking analysis, we found strong predicted binding between CD14 and FimH through specific amino acids in both proteins. Interestingly, the respective FimH amino acids are highly conserved not only among pathogenic, but also non-pathogenic *E. coli* strains. However, although we found FimH to be necessary, we did not find the simple presence of pathogenic *fimH* gene in an otherwise non-pathogenic genetic background to be sufficient for suppression of the immune response. Additionally, the quantitative effects on suppression of the immune response varied strongly between several different clinical isolates. Generally, the pathogenic potential of UPECs does not seem to be the result of a single defined virulence gene, but rather a combination of effects by several factors (*Touchon et al., 2009*). For example, the outer membrane of different clinical isolates displays a wide diversity in protein composition (*Wurpel et al., 2015*) and the production of outer membrane vesicles delivering associated proteins was linked to pathogenesis (*Schwechheimer and Kuehn, 2015*). Therefore, it remains unknown which other pathogenic gene(s) work together with *fimH*, as the UPEC strain CFT073 carries accessory genes encoded by 13 pathogenicity islands (*Lloyd et al., 2007*).

Given how pathogenic bacteria can circumvent the host immune response by generating protected niches (*Grant and Hung, 2013*) and due to the constantly growing presence of multidrug-resistant strains (*Boucher et al., 2009*), treatment of persistent or recurring infections is increasingly challenging. Although vaccination approaches against FimH (*Eldridge et al., 2021*) or non-conventional treatments, such as small mannosides (*Schönemann et al., 2019*; *Mydock-McGrane et al., 2016*), showed promising results, treatments of infections caused by type 1 piliated pathogens remains difficult. Our findings underscore the importance of the mannose-binding domain, and several conserved amino acid residues in this domain, in the interaction between FimH and CD14. For example, R98, showing the strongest binding to CD14 in our predictions, was identified to be important to stabilize the protein–ligand interaction (*Han et al., 2010*). However, targeting this residue only was not sufficient to boost affinity of FimH antagonists (*Tomašič et al., 2021*). Given our results, we hypothesize that this could be because not individual amino acids, but the overall secondary structure of the mannose-binding domain of FimH is important for the interaction with host receptors. Blocking CD14 antibodies, such as the human anti-CD14 antibody IC14 (*Axtelle and Pribble, 2001*), could represent an alternative way to treat recurring infections caused by type 1 piliated pathogens, such as UTIs or inflammatory bowel diseases like Crohn's disease (*Sivignon et al., 2017*), given that CD14 is expressed by hematopoietic but also non-hematopoietic cells (*Zanoni and Granucci, 2013*).

Although opportunistic pathogens continuously circulate in humans as asymptomatic colonizers, they occasionally cause symptomatic acute or even chronic infections in some individuals (*Donnenberg, 2013*). Opportunistic pathogens use type 1 pili to adhere to and invade into host epithelial cells. While the cellular invasion represents a *de facto* passive mechanism for the pathogens to hide from the immune system and to establish persistent or recurring infections, we found that UPECs use type 1 pili to actively manipulate the behavior of innate immune cells, and thus also the adaptive immune response, by direct interaction of FimH and CD14 receptor on DCs. Binding of FimH to CD14 overactivated the NFAT pathway leading to expression of immunomodulatory cytokines. For example, we found increased levels of IL-10 which acts as an immunosuppressant decreasing the expression of co-stimulatory molecules on DCs (*Mittal and Roche, 2015*), drives differentiation of regulatory T cells (*Hsu et al., 2015*) and is a hallmark cytokine produced by tolerogenic DCs (*Domogalla et al., 2017*). Although the immunosuppressive effects that type 1 piliated UPECs exert on DCs after the FimH–CD14 interaction are averaged in larger populations by non-cell autonomous effects during mixed infections, our results underscore the importance of the binary response during pathogen–host interaction at the single-cell level given that early immune response to UPECs can decide on the recurrent fate of the infection (*Hannan et al., 2010*). Moreover, since CD14 is a multifunctional co-receptor of several immune cell types (*Zanoni and Granucci, 2013*) and type 1 pili are expressed not only by pathogenic but also by non-pathogenic *E. coli* (*Shawki and McCole, 2017*; *Croxen and Finlay, 1999*), our findings add a new layer of complexity to the physiological relevance of type 1 pili – as modulators of the immune response in general and specifically during persistent and recurring infections.

## Methods

### Experimental model and subject details

#### Animals

Mice were bred and maintained at the local animal facility in accordance with the IST Austria ethics commission. All experiments were conducted in accordance with the Austrian law for animal experiments. Permission (BMWFW-66.018/0010-WF/V/3b/2016) was granted by the Austrian federal ministry of science, research, and economy.

#### Cell culture

R10 medium, RPMI 1640 + 10% FCS (fetal calf serum), 2 mM L-glutamine, 50 µM beta-mercaptoethanol, 100 U/ml penicillin, and 100 µg/ml streptomycin, was used as basic medium. Stem cell medium was supplemented with 10 ng/ml IL-3, 20 ng/ml IL-6% and 1% stem cell factor (SCF) supernatant produced by B16 melanoma cells. Hoxb8 medium was supplemented with 125 ng/ml Flt3 and 1 µM estradiol. iCD medium was supplemented with 2 ng/ml GM-CSF and 75 ng/ml Flt3. GM-CSF and Flt3 supernatants were produced by hybridoma cells and concentration of cytokines was measured by enzyme-linked immunosorbent assay (ELISA). All media were used pre-warmed. Cells were grown routinely at 37°C with 5% $CO_2$. For infection and subsequent assays, cells were cultured in R10 medium without antibiotics and buffered with 20 mM HEPES (4-(2-hydroxyethyl)-1-piperazineethanesulfonic acid) (R10H20) in the absence of $CO_2$.

#### WT, $Itgb2^{-/-}$, $Tlr4^{-/-}$, and $Cd14^{-/-}$ Hoxb8 cell generation

Five-week-old WT C57BL/6J, $Itgb2^{-/-}$ (B6.129S7-$Itgb2^{tm1Bay}$/J), $Tlr4^{-/-}$ (B6(Cg)-$Tlr4^{tm1.2Karp}$/J), and $Cd14^{-/-}$ (B6.129S4-$Cd14^{tm1Frm}$/J) mice were obtained from the Jackson Laboratory. Immortalization of bone marrow cells was performed as described previously (*Leithner et al., 2018*; *Redecke et al., 2013*). In brief, bone marrow was isolated from the femur and tibia by centrifugation and cells were precultured in stem cell medium for 3 days to enter the cell cycle. $1 \times 10^5$ cells were spin infected with Hoxb8-MSCV retrovirus using lipofectamine at $1000 \times g$ for 1 hr in Hoxb8 medium. Cells were fed and split every few days for 3–4 weeks until all uninfected cells died off and only Hoxb8 infected, immortalized, cells were left.

#### DC differentiation

iCD103 DCs, expressing CD103, were differentiated from Hoxb8 progenitor cells as described previously for bone marrow cells with minor modifications (*Mayer et al., 2014*). In brief, Hoxb8 cells were seeded at a density of $3 \times 10^5$ cells into 10-cm bacterial culture dishes in 10-ml iCD medium. On day 3, cells were split 1:2 and toped up with fresh iCD medium to 10 ml. On day 6, cells were fed with 10-ml iCD medium and non-adherent iCD103 DCs were frozen on day 9. For images of immature and matured cells, as well as flow cytometry staining of different surface markers see *Figure 1—figure supplement 1*.

Frozen DCs were allowed to recover after thawing for at least 4 hr before infection. Non-adherent cells were counted and seeded at a density of $1–2 \times 10^5$/ml in R10H20 medium. Assays were performed with DCs that were either stimulated with bacteria or recombinant LPS at 200 ng/ml. $Itgb2^{-/-}$ DCs were purified from potential other cells using the EasySep mouse Pan-DC enrichment kit (Stemcell Technologies) and allowed to rest for 1 hr before the infection assay.

#### Bacterial strain construction

For cloning, strains were grown routinely in LB medium. Plasmids were maintained at 100 µg/ml ampicillin or 50 µg/ml kanamycin. Single-copy integration was performed at 12 µg/ml chloramphenicol or 25 µg/ml kanamycin. For experiments, strains were grown at 37°C in R10H20 medium without antibiotics or in medium containing 0.5% casamino acids, 1× M9 salts, 1 mM $MgSO_4$, 0.1 mM $CaCl_2$, and 0.5% glycerol (CAA M9 glycerol). Primers used for cloning are listed in *Table 1* and strains used are listed in *Table 2*.

**Table 1.** Primers used for cloning.

| Oligonucleotide | Sequence |
| --- | --- |
| 110 | ATATGCATGCCAAAAGATGAAACATATATCATAAATAAGTTACGT |
| 112 | ATATGCATGCCAAAAGATGAAACATTCATAGAGGAAAGCATCG |
| 119 | CAGTAATGCTGCTCGTTTTGCCG |
| 120 | GACAGAGCCGACAGAACAAC |
| 128 | TGTGTAGGCTGGAGCTGCTTC |
| 130 | AAAAGAGAAGAGGTTTGATTTAACTTATTGATAATAAAGTTAAAAAAACACTGCTTCGAAGTTCC |
| 131 | CACTTTGTTTTGTAAACGAGTTTGACTGCCAACACTGCACAGTTTTCCCCCAAAAGATGAAACAT |
| 132 | ATTCATATGGAATAAATACAAGACAATCATAGAGGAAAGCATC |
| 133 | ATTCATATGGAATAAATACAAGACAAATATCATAAATAAGTTACGTATTTTTTCTCAAGCATAAAAATATTAAAAAACGAC |
| 134 | TTGTATTTATTCCATATGAATATCCTCCTTAGTTCCTATTCC |
| 146 | TAGCTTCAGGTAATATTGCGTACCTGCATTAGCAATGCCCTGTGATTTCTCCATATGAATATCCTCCTTAGTTCC |
| 148 | AGTGATTAGCATCACCTATACCTACAGCTGAACCCAAAGAGATGATTGTATGTGTAGGCTGGAGCTGCTTCG |
| 157 | AGCATCACCTATACCTACAGCTG |
| 158 | AGCTTCAGGTAATATTGCGTACC |
| 198 | AGTGATTAGCATCACCTATACCTACAGCTGAACCCGAAGAGATGATTGTAATGAAACGAGTTATTACC |
| 276 | TAGCTTCAGGTAATATTGCGTACCTGCATTAGCAATGCCCTGTGATTTCTTTATTGATAAACAAA |
| 296 | AGTGATTAGCATCACCTATACCTACAGCTGAACCCGAAGAGATGATTGTATTGACGGCTAGCTCAGTCCTAGGTA |
| 297 | TAGCTTCAGGTAATATTGCGTACCTGCATTAGCAATGCCCTGTGATTTCTTCAGCACTGTCCTGCTCCTTGTGAT |
| 3_SphI_pKD3_test | TGAATACCACGACGATTTCC |
| cam_test_R | CAACGGTGGTATATCCAGTGA |
| FarChro galK UO | GTTAATTATCATTTTGCACCGCGTC |
| galK-KpnI-r | CCGGGTACCTCAGCACTGTCCTGCTCC |
| galK-ver-F | CCTACTCTATGGGCTGGCAC |
| galk-ver-R | GGAAAGTAAAGTCGCACCCC |

## Locked mutants

*E. coli* W (**Archer et al., 2011**; ATCC 9637) and clinical isolates CFT073 (**Welch et al., 2002**; ATCC 7000928; a kind gift of Ulrich Dobrindt), UTI89 (**Chen et al., 2006**), and 536 (**Hochhut et al., 2005**) were used as well as derivatives of those strains.

Phase-locked mutants were generated by replacing the 9-bp long recognition site for the site-specific recombinases FimB and FimE in the internal repeat region on the left site of the fim switch (*fimS*) with an FRT site (**Figure 1C**). The *fimS* region was amplified either in the ON (primers 132 and 110) or OFF (primers 133 and 112) orientation from the chromosome of the WT strain but omitting the 9-bp recognition site on the left site. An FRT-removable chloramphenicol resistance marker was amplified from pKD3 plasmid using primers 128 and 134 (**Datsenko and Wanner, 1999**). The resistance marker was assembled left of the amplified *fimS* regions using NEBuilder assembly kit (NEB). The assembled DNA fragments were PCR amplified using primers 130 and 131 and integrated into the chromosome of the respective strains instead of the endogenous *fimS* using lambda red recombination (**Datsenko and Wanner, 1999**). In brief, *E. coli* W or CFT073 WT bacteria were transformed with pSIM6 plasmid expressing thermal inducible Red genes under control of the native $\lambda$ phage *pL* (**Datta et al., 2006**) and selected with ampicillin at 30°C. After inducing expression of lambda red genes at 42°C for 15 min, bacteria were made electrocompetent and transformed with 100 ng of the cleaned PCR fragments. Bacteria were allowed to recover in LB medium for 1 hr at 37°C before spreading on LB plates containing chloramphenicol.

**Table 2.** Used strains.

| Strain | Reference |
| --- | --- |
| *E. coli* W (ATCC 9637) | *Archer et al., 2011* |
| CFT073 O6:K2:H1 (ATCC 700928) | *Welch et al., 2002* |
| UTI89 O18:K1:H7 | *Chen et al., 2006* |
| 536 O6:K15:H31 | *Hochhut et al., 2005* |
| KT177 (UTI89 locked-ON::FRT) | This paper |
| KT178 (UTI89 locked-OFF::FRT) | This paper |
| KT179 (CFT073 locked-ON::FRT) | This paper |
| KT180 (CFT073 locked-OFF::FRT) | This paper |
| KT193 (CFT073 locked-ON::FRT Δ*fimH*::FRT) | This paper |
| KT232 (*E. coli* W locked-ON::FRT) | This paper |
| KT257 (CFT073 locked-ON::FRT Δ*fimH* att λ $P_R$-*mVenus*::FRT) | This paper |
| KT260 (CFT073 locked-ON::FRT Δ*galK*::FRT *fimH*::*fimH*$_{Y48A\ R98A\ T99A}$) | This paper |
| KT261 (CFT073 locked-ON::FRT Δ*galK*::FRT *fimH*::*fimH*$_{Y48A}$) | This paper |
| KT262 (CFT073 locked-ON::FRT Δ*galK*::FRT *fimH*::*fimH*$_{R98A}$) | This paper |
| KT263 (CFT073 locked-ON::FRT Δ*galK*::FRT *fimH*::*fimH*$_{T99A}$) | This paper |
| MG002 (*E. coli* W locked-ON::FRT Δ*galK*::FRT *fimH*::*fimH*$_{CFT073}$) | This paper |
| VG003 (CFT073 locked-ON::FRT att*λ* $P_R$-*mVenus*::FRT) | This paper |
| KT280 (536 locked-ON::FRT) | This paper |
| KT281 (536 locked-OFF::FRT) | This paper |

After verifying single-copy integration using primers 119, 120, cam_test_R and 3_SphI_pKD3_test, the resistance marker was subsequently removed using pCP20 plasmid (*Cherepanov and Wacker-nagel, 1995*). The mutated *fimS* region was sequenced to confirm deletion of the 9-bp long recombinase recognition site on the left site, but a fully intact recognition site on the right site. The presence or absence of the type 1 pilus on the bacterial outer membrane was confirmed by electron microscopy and yeast agglutination assay. Resulting locked mutants were: CFT073 locked-ON – KT179, CFT073 locked-OFF – KT180, and *E. coli* W locked-ON – KT232.

## FimH deletion mutant

*fimH* gene from CFT073 locked-ON mutants (KT179) was deleted by lambda red recombination. The FRT-flanked chloramphenicol resistance marker from pKD3 plasmid was amplified using primers 146 and 148 and integrated into the chromosome of KT179 strain. Successful deletion was confirmed by PCR (primers 157 and 158). Resistance was flipped using pCP20 plasmid resulting in CFT073 locked-ON Δ*fimH* mutant – KT193. The presence of type 1 pili was confirmed by electron microscopy. The absence of *fimH* was confirmed by sequencing and yeast agglutination assay.

## Chromosomal yfp marker

*mVenus* driven by the right site of the lambda $P_O$ was integrated in the lambda phage attachment site on the chromosome of CFT073 locked-ON (KT179) and CFT073 locked-ON Δ*fimH* (KT193) mutants using CRIM integration (*Haldimann and Wanner, 2001*). In brief, KT179 and KT193 strains were transformed with pInt-ts helper plasmid and selected on ampicillin plates at 30°C. Bacteria were made electrocompetent and transformed with $P_R$-*mVenus* carrying pAH120-frt-cat integration plasmid. After recovery in LB medium for 1 hr at 37°C, the expression of lambda red genes was induced at 42°C for 15 min before spreading on LB plates containing chloramphenicol.

Single-copy integration of the CRIM plasmid was verified with PCR as mentioned previously (*Haldimann and Wanner, 2001*). Since pAH120-frt-cat was designed to have an FRT-flanked chloramphenicol resistance marker (*Nikolic et al., 2018*), resistance was subsequently removed using pCP20 plasmid (*Cherepanov and Wackernagel, 1995*). Resulting mutants were: CFT073 locked-ON *attλ* $P_R$-mVenus – VG003 and CFT073 locked-ON Δ*fimH attλ* $P_R$-mVenus – KT257.

## FimH replacement mutant

The endogenous *fimH* gene of the non-pathogenic *E. coli* W strain was exchanged scar-less with *fimH* of the pathogenic UPEC strain CFT073 using *galK* selection/counter-selection (*Kavčič et al., 2020*). The FRT-flanked kanamycin resistance marker from gDNA harboring Δ*galK::kan* (gift of Bor Kavčič) was amplified using primers galK-ver-F and galK-ver-R (gift of Bor Kavčič) and integrated into the *galK* gene of *E. coli* W locked-ON (KT232). Loss of *galK* gene was confirmed by PCR using primers FarChro galK UO and galK-KpnI-r (gift of Bor Kavčič). Resistance was flipped using pCP20 plasmid. *galK* under constitutive J23100 promoter was amplified from pKD13-PcgalK plasmid using primers 296 and 297 and transformed to replace the endogenous *fimH* gene using lambda red recombination. After recovery, any residual carbohydrate residues were removed by washing the cells several times with M9 buffer (*Tomasek et al., 2018*) before plating on M9 minimal medium containing 0.1% galactose as only carbohydrate source for positive selection. Integration of *galK* gene into *fimH* was confirmed by PCR using primers 157 and 158. CFT073 *fimH* gene was amplified from a gblock (IDT) carrying the *fimH* sequence from CFT073 using primers 198 and 276 and integrated instead of the constitutive *galK* gene. After recovery, cells were washed several times as before. Transformants were counter-selected on artificial urine medium agar plates (AU-Siriraj; *Chutipongtanate and Thongboonkerd, 1999*) supplemented with 20 µg/ml L-aspartate and 20 µg/ml L-isoleucine (*Bouvet et al., 2017*), and containing 0.2% 2-deoxy-D-galactose (DOG) and 0.2% glycerol for the counter-selection. Pathogenic *fimH* integration was confirmed by PCR using primers 157 and 158. Resulting mutant was *E. coli* W locked-ON *fimH::fimH_{CFT073}* – MG002.

## FimH amino acid mutants

Single and triple point mutants of amino acids predicted to be most involved in binding to CD14 were generated using *galK* selection/counter-selection as mentioned above. Briefly, *galK* was deleted from CFT073 locked-ON mutants (KT179). Thereafter constitutive expressed *galK* was inserted in the endogenous *fimH* of this strain for selection. 100 µg gblocks (IDT) carrying either Y48A, R98A, T99A or the triple mutation (Y48A, R98A, T99A) in the *fimH* sequence from strain CFT073 were integrated instead of the constitutive *galK* gene. Correct integration of *fimH* having mutated amino acid residues was confirmed by sequencing. Resulting mutants were CFT073 locked-ON *fimH_{Y48A R98A T99A}* – KT260, CFT073 locked-ON *fimH_{Y48A}* – KT261, CFT073 locked-ON *fimH_{R98A}* – KT262, and CFT073 locked-ON *fimH_{T99A}* – KT263.

## Inhibitors used

Since the suggested upper daily limit of orally applied D-mannose to treat UTIs is 9 g, leading to blood mannose levels of roughly 175 µM (*Alton et al., 1997*), we decided to compare this concentration to a strongly increased one of 30–60 g D-mannose resulting in 1 mM blood mannose levels. The small mannoside M4284 (medchemexpress) was used at 10 µM. The blocking CD14 antibody M14-23 was used at 20 µg/ml (*Tsukamoto et al., 2010*). Calcineurin inhibitor FK506 was used at 50 nM. DCs were pre-incubated with FK506 20 min prior to bacterial stimulation, whereas all other inhibitors were added as the same time as the bacteria.

## Method details

### Yeast agglutination assay

*Saccharomyces cerevisiae* was grown in YPD medium at 30°C for 2 days. After centrifugation, cells were resuspended in M9 buffer to an $OD_{600}$ of 1 and stored in the fridge. An aliquot of the yeast was transferred to glass slides and bacterial colonies were directly mixed into the yeast solution. Agglutination occurred within few seconds to 1 min. Pictures of agglutinated yeast and bacteria cells were taken on a brightfield microscope at ×10 magnification and images were processed with Fiji.

## Growth curve assay and doubling time estimation

Single bacterial colonies were inoculated in 160 µl R10H20 in 96-well plates and grown overnight at 220 rpm at 37°C. The next day, cultures were diluted 1 in 1000 in R10H20 supplemented with 0.0005% Triton-X and grown at 37°C with shaking. Optical density was measured every 30 min at 600 nm at a Synergy H1 plate reader for a total of 7 hr. The data were blank normalized, and the doubling time ($dt$) was calculated from exponential data using following formula $dt = \frac{t2-t1}{\left(3.3*\log\left(\frac{OD_{600}2}{OD_{600}1}\right)\right)}$ .

For all assays, bacteria were grown to early- to mid-exponential phase ($OD_{600}$ 0.25; *E. coli* W 4 hr, CFT073 3 hr 20 min; see *Figure 1—figure supplement 2A*) in 1 ml R10H20 medium, if not indicated otherwise, at 37°C in deep well plates.

## Minimal inhibitory concentration assay

Cultures were grown to $OD_{600}$ of 0.25 and approximately $10^6$ bacteria were used for MIC assays. Serial dilutions of gentamicin were performed in a microdilution manner using 96-well plates and $OD_{600}$ was measured after 18 hr incubation at 37°C with shaking. The threshold to calculate the minimal inhibitory concentration (MIC) was set to detectable growth above the blank background after normalization. CFT073 locked-ON (KT179) and locked-OFF (KT180) mutants had similar MIC to gentamicin in R10H20 medium (see *Figure 1—figure supplement 2B*). 5× the MIC was used to prevent extracellular growth of bacteria in the infection assays (7.5 µg/ml).

## Electron microscopy

Bacteria were grown to mid-exponential phase in CAA M9 glycerol medium, fixed with glutaraldehyde (EM grade, final concentration 2.5 %) for 30 min at 4°C, washed twice with phosphate-buffered saline (PBS) and concentrated in water. Formvar-coated copper grids were glow discharged for 2 min and $5 \times 10^6$ bacteria were loaded for 5 min. Excess liquid was removed with filter papers and bacteria were stained with Uranyless for 2 min. After removal of excess liquid, the grids were washed 10 times in water and dried. EM images were taken at TEM T10 microscope at 80 kV. Images were processed with Fiji.

## Predicted protein–protein interaction

Crystal structures of FimH (PDB: 6GTV), mouse CD14 (PDB: 1WWL), human CD14 (PDB: 4GLP), mouse TLR4 (PDB: 3VQ2), and mouse CD48 (PDB: 2PTV) were obtained from rscb.org, cleaned from solvents and other co-precipitated molecules using PyMol and run on HawkDock server to predict protein binding (*Weng et al., 2019*). Additional MM/GBSA (molecular mechanics energies/generalized Born and surface area continuum solvation) analysis was run to predict free binding energy (*Hou et al., 2005*).

## Bead-binding assay

5 µl of 10 µm bead slurry (PureProteom Protein A magnetic beads, Merck) were used per reaction. Beads were washed 3× with PBS containing 0.005% Tween (PBS-T). Beads were collected using a magnetic stand. 5 µg CD14-Fc recombinant protein (RND System) in a total of 10 µl of PBS-T was coupled to the beads for 1 hr at 4°C with rotation. After washing the beads, 100 µl of bacteria grown to early- to mid-exponential phase in CAA M9 glycerol medium (~$1 \times 10^7$ bacteria) were incubated in the presence of Tween with the CD14-coupled beads for 1 hr at 4°C with rotation. After washing, this time with M9 glycerol medium containing Tween, bead-bound bacteria were fixed with 0.5% PFA (paraformaldehyde) in M9 buffer for 10 min at 4°C.

For flow cytometry, samples were diluted in M9 buffer and analyzed on a FACS Canto II (BD). 10,000 events gated on FSC-A and SSC-A to exclude debris were recorded at medium flow rate and data were analyzed using FlowJo software. Three things should be noted: First, we observed that due to the high force applied in the sample injection tube during acquisition, the bacteria were separated from the magnetic beads they previously were bound to, leading to a single detectable fluorescent peak only. Second, due to their size, the magnetic beads are also detected in the gating range specific for the bacterial population (see *Figure 4—figure supplement 2*). Third, as can also be seen in the fluorescent images, the magnetic beads have weak auto-fluorescence in the FITC channel.

We therefore quantified the amount of fluorescent events applying the same gating strategy as for the bacterial population only (*Figure 4—figure supplement 2* and C).

For fluorescent microscopy, samples were embedded in mounting buffer and spread on a cover slip. Images were taken with 100× magnification on a custom-built Olympus widefield microscope with Hamamatsu Orca Flash4.0v2 camera and LED-based fluorescence illuminator using YFP (x513/22,m543/22) fluorescence channel (*Chait et al., 2017*). Images were processed with Fiji and deconvoluted with Huygens software.

## Type 1 pili extracts

Type 1 pili extracts were generated as described previously (*Sheikh et al., 2017*) with minor modifications. Briefly, CFT073 locked-ON (KT179) and CFT073 locked-ON Δ*fimH* (KT193) were grown overnight in CAA M9 glycerol medium and harvested at 4000 × *g* for 1 hr. The cell pellet was resuspended in 1 mM Tris–HCl (pH 8.0) and incubated at 65°C for 1 hr with occasional vortexing. After pelleting the cells at 15,000 × *g* for 10 min, type 1 pili were precipitated from the supernatant overnight in the presence of 300 mM NaCl and 100 mM MgCl$_2$ at 4°C. Type 1 pili were concentrated at 20,000 × *g* for 10 min, washed once with 1 mM Tris–HCl and snap frozen in a small volume of 1 mM Tris–HCl.

## Dot blot assay

The recombinant chimera CD14-Fc protein was biotinylated using the EZ-Link Sulfo NHS-LC-LC-Biotin kit (Thermo Fisher) and a fivefold molar excess of biotin. In brief, CD14-Fc was dissolved in PBS at 1 mg/ml and biotinylated with a fivefold molecular excess of biotin for 30 min at room temperature (RT). The biotinylation reaction was stopped with 3 mM Tris–HCl (pH 7.0).

The PVDF (polyvinylidene difluoride) membrane was activated for 5 min with MeOH, washed for 5 min in water and allowed to dry for 5 min. 20 ng biotinylated CD14 and roughly 20 µg type 1 pili extract from CFT073 ON (KT179) and CFT073 ON Δ*fimH* (KT193) mutants were loaded onto the membrane. Protein spots were allowed to dry for 10 min and then the membrane was blocked in 3% bovine serum albumin (BSA) in PBS for 30 min at 37°C. 40 ng biotinylated CD14 in 3% BSA in PBS and 0.05% Tween was blotted on the membrane and incubated for 1 hr at 37°C. Then the membrane was washed 3× in PBS with Tween for 5 min each. Streptavidin–HRP antibody was pre-diluted 1:100 in 3% BSA in PBS with Tween and diluted once more 1:5000 in PBS with Tween. The membrane was incubated with streptavidin–HRP for 1 hr at RT. After washing again 3× as before, chemiluminescence was detected using clarity ECL substrates (BioRad).

## Infection assays

DCs were seeded at a density of 1–2 × 10$^5$ cells/ml in R10H20 medium (for adhesion assays black 24-well tissue-treated dishes were used, for any other assay non-treated dishes were used). DCs were matured either by addition of 200 ng/ml recombinant LPS or bacteria at a multiplicity of infection of 10 (10 bacteria per 1 DC). 1 hr post infection (pi) gentamicin was added at 7.5 µg/ml to prevent extracellular growth of bacteria. 18–20 hr pi subsequent assays were performed.

## Cytokine array

Differences in secreted cytokines of WT, WT pretreated for 20 min with 50 nM FK506 or *Cd14*$^{-/-}$ DCs after CFT073 locked-ON (KT179) or OFF (KT180) stimulation were analyzed using a membrane-based antibody array (Mouse Cytokine Array Kit, Abcam). The assay was carried out according to the manufacturer's protocol using 1 ml of cell culture supernatant and incubating overnight. Array images were processed with Fiji and data were normalized to the positive controls of WT ON replicate one following the manufacturer's recommendations for comparison.

## ELISA

Secreted interferon-alpha 1 of WT, WT pretreated for 20 min with 50 nM FK506 or *Cd14*$^{-/-}$ DCs after CFT073 locked-ON (KT179) or OFF (KT180) stimulation was analyzed using ELISA (Mouse Interferon-alpha 1 ELISA, Abcam). The assay was carried out according to the manufacturers protocol using undiluted cell culture supernatant.

## Adhesion assay

Non-adherent DCs were removed, and adherent cells were washed twice with 500 µl PBS. Adherent cells were stained with Hoechst 33,342 (NucBlue reagent, 2 drops/ml) in R10H20 medium for 30 min at 37°C. Cells were washed twice and 1 ml Live Cell Imaging solution (140 mM NaCl, 2.5 mM KCl, 1.8 mM CaCl$_2$, 1.0 mM MgCl$_2$, 20 mM HEPES, pH 7.4) was added to the wells. Fluorescence was measured with a Synergy H1 plate reader (excitation 490 nm, emission 520 nm, bottom reading without lid, 50 data points per well). Pictures of adherent cells were taken on a brightfield microscope at ×4 magnification and images were processed with Fiji.

## Flow cytometry staining

DCs were collected and incubated in FACS (fluorescence activated cell sorting) buffer (1× PBS, 2 mM EDTA (ethylenediaminetetraacetic acid), 1% BSA; RT) or Tyrode's buffer (used for active CD11b and β1 staining, on ice) with Fc receptor block for 20 min. Cells were stained for 30 min with antibodies using the respective buffer with Fc receptor block. Cells were washed twice with PBS and resuspended in the respective buffer for analysis on FACS Canto II (BD). 10,000 events gated on FSC-A and SSC-A to exclude debris were recorded at medium flow rate. Data were analyzed using FlowJo software by performing doublet discrimination. Antibodies used are listed in *Table 3*.

## Receptor endocytosis assay

Surface staining of receptors was performed as described previously (*Zanoni et al., 2011*). DCs were fixed for 10 min at RT by adding PFA to a final concentration of 2% at the time points indicated. Fixed DCs were collected and resuspended in FACS buffer and stained for 30 min with antibodies. After washing, cells were analyzed as mentioned above.

## Ex vivo assays

### Ear crawl-out

Ear crawl-out assays were performed similar as published previously (*Kopf et al., 2020*). In brief, ears from 5-week-old female C57Bl/6J WT mice were first UV sterilized for 10 min and then split into dorsal and ventral halves. Ventral halves were placed in R10H20 medium, ventricles facing down. Ears were incubated with 10$^6$ CFT073 locked-ON (KT179) or OFF (KT180) bacteria for 48 hr, renewing the infection stimulus after 24 hr. 1 hr after every infection 7.5 µg/ml gentamicin was added to the medium. Ears were fixed using 4% PFA and immersed using 0.2% Triton-X. After blocking in 1% BSA in PBS, lymphatics were stained for 90 min using rat anti-Lyve-1 antibody and DCs were stained using biotinylated anti-MHCII antibody. Secondary antibodies, anti-rat F(ab')$_2$-AF488 and Streptavidin-AF647, were used subsequently for 45 min each. Ears were mounted on cover slips with ventricles facing up using cover glasses. 10 µm z-stacks were taken on inverted LSM800 confocal microscope with 488 and 640 nm LED-laser light source. Images were taken from three biological replicates (except *Tlr4$^{-/-}$* where two biological replicates were imaged) analyzing at least two field of views each. Maximum intensity projection images were processed with Fiji. Images were analyzed using custom-made scripts in Fiji. Pre-processing was done using lymphatics script and analysis using LVmeanDCarea script.

### Ear crawl-in

Ear crawl-in assays were performed similar as published previously (*Leithner et al., 2016*). In brief, ears from 5-week-old female C57BL/6J mice were split as described above. WT DCs stimulated with CFT073 locked-ON (KT179) or OFF (KT180) were labeled with 10 µM TAMRA or 3 µM Oregon green, respectively, or vice versa, and either used individually or mixed at a 1:1 ratio. 6 × 10$^4$ cells were allowed to invade the ear tissue for 30 min. Non-invaded cells were washed off and ears were incubated at 37°C for 6 hr. Ears were fixed and lymphatics were stained using rat anti-Lyve-1 and anti-rat F(ab')$_2$-AF647 antibody. Ears were mounted on cover slips and imaged as described above.

### In vitro 2D migration assay

After performing infection assays in 24-well tissue-treated dishes, non-adherent DCs were removed and adherent DCs were gently washed with PBS. 1 ml of fresh R10 medium containing uniform CCL19 chemokine (RND Systems) (0.625 µg/ml) was added per well. Images were taken every 30 s for a total

**Table 3.** Antibodies used.

| Antibody | Source | Identifier |
|---|---|---|
| Rat anti-MHC II (I-A/I-E) eFluor450 (clone M5/114.15.2) | eBioscience | 48-5321-82 |
| Armenian hamster anti-CD11c APC (clone N418) | eBioscience | 17-0114-82 |
| Rat anti-CD18 PE (clone C71/16) | BD | 553,293 |
| Rat anti-CD11b FITC (clone M1/70) | eBioscience | 11-0112-82 |
| Mouse anti-human CD11b (active epitope) APC (clone CBRM1/5) | eBioscience | 17-0113-41 |
| Rat anti-CD86 biotin (clone GL1) | eBioscience | 13-0862-85 |
| Armenian hamster anti-CD80 (clone 16–10 A1) | eBioscience | 13-0801-85 |
| Rat anti-CD40 (clone 1C10) | Biolegend | 102,802 |
| Armenian Hamster anti-CD103 Brilliant Violet 421 (clone 2E7) | Biolegend | 121,421 |
| Rat anti-CD14 Brilliant Violet 421 (clone Sa14-2) | Biolegend | 123,329 |
| Rabbit anti-CD14 (clone EPR21847) | Abcam | ab221678 |
| Mouse anti-CD64 (FcγRI) Brilliant Violet 421 (clone X54-5/7.1) | Biolegend | 139,309 |
| Rat anti-CD4 eFluor450 (clone GK1.5) | eBioscience | 48-0042-82 |
| Armenian Hamster anti-CD69 APC-eFluor780 (clone H1.2F3) | eBioscience | 47-0691-82 |
| Rat anti-CD62-L PE (clone MEL-14) | eBioscience | 12-0621-82 |
| Mouse anti-CD45.2 APC (clone 104) | eBioscience | 17-0454-82 |
| Rat IgG2b kappa Isotype Control eFluor450 (clone eB146/10H5) | eBioscience | 48-4031-82 |
| Armenian Hamster IgG Isotype Control APC (clone eBio299Arm) | eBioscience | 14-4888-81 |
| Rat IgG2a kappa Isotype Control PE (clone eBR2a) | eBioscience | 12-4321-81 |
| Rat IgG2b kappa Isotype Control FITC (clone eB149/10H5) | eBioscience | 11-4031-82 |
| Rat IgG1 kappa Isotype Control APC (clone eBRG1) | eBioscience | 17-4301-82 |
| Armenian Hamster IgG Brilliant Violet 421 (clone HTK888) | Biolegend | 400,936 |
| Rat IgG2a kappa Isotype Control Biotin (clone RTK2758) | Stemcell Technologies | 60,076 |
| Armenian Hamster IgG Isotype Control Biotin (clone eBio299Arm) | eBioscience | 13-4888-81 |
| Rat IgG2a kappa Isotype Control (clone RTK2758) | Stemcell Technologies | 60,076 |
| Alexa Fluor 647-conjugated Streptavidin | Jackson ImmunoResearch | 016-600-084 |
| Donkey anti-rat IgG H + L Alexa Fluor488 AffiniPure F(ab')2 Fragment | Jackson ImmunoResearch | 712-546-150 |
| Donkey anti-rat IgG H + L Alexa Fluor647 AffiniPure F(ab')2 Fragment | Jackson ImmunoResearch | 712-606-150 |
| Goat anti-rabbit Alexa Fluor488 | Invitrogen | A11008 |
| Rat anti-CD29 (clone 9EG7) | BD | 553,715 |
| Armenian Hamster anti-CD29 PE (clone HMβ1–1) | Biolegend | 102,207 |
| Armenian Hamster IgG Isotype Control PE (clone HTK888) | Biolegend | 400,907 |
| M14-23 (anti-mouse CD14 antibody) | Biolegend | 150,102 |
| Strep-Tactin HRP | iba-lifesciences | 2-1502-001 |
| Fc receptor block | eBioscience | 14-9161-73 |

of 5 hr on bright field microscopes using ×10 magnification and an exposure of 20ms. Data were analyzed using a custom-made R script: Tracking_migration_single_cell script.

## In vitro 3D collagen migration assay

3D collagen chemotaxis assays were performed as described previously (*Leithner et al., 2016*), with minor modifications. Assays were performed in PureCol bovine collagen with a final collagen concentration of 1.6 mg/ml in 1× minimum essential medium eagle (MEM) and 0.4% sodium bicarbonate using 1–2 × 10⁵ DCs. The collagen–cell mixture was cast to custom-made migration chambers and polymerized for 1 hr at 37°C. CCL19 chemokine (RND Systems) in R10 (0.625 µg/ml) was pipetted on top of the gel and the chambers were sealed with paraffin. Images were taken every 30 s for a total of 5 hr on bright field microscopes using ×4 magnification and an exposure of 20ms. Data were analyzed using custom-made Fiji scripts: images were pre-processed using Tracking_pre-processing_for_brightfield script and analyzed using migrationspeedREP script.

## In vitro extracellular matrix migration assay

Cell-derived matrixes (CDMs) were produced as described previously (*Kaukonen et al., 2017*). In brief, round shaped coverslips were coated with 0.2% gelatin in PBS in 24-well dishes for 1 hr at 37°C. Gelatin was crosslinked with 1% glutaraldehyde in PBS for 30 min at RT and quenched with 1 M glycine in PBS for 20 min at RT. After washing the coverslips twice with PBS, 5 × 10⁴ 3T3 mouse fibroblasts in DMEM (Dulbecco's Modified Eagle Medium), GlutaMAX, supplemented with 10% FCS, 100 U/ml penicillin and 100 µg/ml streptomycin were seeded per well. After 48 hr 3T3 fibroblasts reached confluency and were treated daily with ascorbic acid for better crosslinking of the extracellular matrix. Old medium was gently removed and fresh medium with 50 µg/ml sterile ascorbic acid was added for 10–14 days. 3T3 fibroblasts were extracted with extraction buffer (0.5% Triton-X, 20 mM $NH_4OH$ in PBS) for 2 min and washed twice with PBS containing 1 mM $CaCl_2$ and 1 mM $MgCl_2$ (PBS/Ca/Mg). DNA was digested with 100 µg/ml DNaseI in PBS/Ca/Mg for 1 hr at 37°C and CDMs were washed twice with PBS/Ca/Mg before storage in PBS/Ca/Mg supplemented with 100 U/ml penicillin and 100 µg/ml streptomycin at 4°C.

Before use, CDMs were placed onto custom-made imaging chambers and incubated with R10 medium for 1 hr at 37°C. The medium was removed and a 1 µl CCL21 chemokine (RND Systems; 25 µg/ml) spot was injected into the CDM and incubated for 10 min. 1 ml R10 medium was added on top and the CDM was incubated for 1 hr at 37°C to allow a chemokine gradient to form. After washing twice gently with R10 medium, CFT073 locked-ON (KT179) and locked-OFF (KT180) stimulated DCs were concentrated by centrifugation and the dense cell pellet was pipetted into the CDM at the opposite site to the chemokine spot. 2 ml of R10 medium was added and images were taken every minute for a total of 6 hr on a bright field microscope using ×10 magnification and an exposure of 20ms. Single cells outside of the cell cluster were counted after 5 hr. Images were processed with Fiji.

## In vivo migration

In vivo migration assays were performed similar as published previously (*Leithner et al., 2018*). In brief, DCs stimulated with CFT073 locked-ON (KT179) or OFF (KT180) were labeled with 10 µM TAMRA or 3 µM Oregon green, respectively, and vice versa. In total, 10⁶ cells – either ON or OFF stimulated DCs separately or ON and OFF stimulated DCs mixed in a 1:1 ratio – were suspended in 25 µl PBS and injected subcutaneously into the hind footpad of 4- to 6-week-old C57BL/6J or Pep Boy (B6 CD45.1, B6.SJL-Ptprca Pepcb/BoyJ) mice. 48 hr later, mice were sacrificed and the popliteal lymph nodes were collected. Lymph nodes were ripped open and digested for 30 min at 37°C in complete DMEM supplemented with 2% FCS, 100 U/ml Penicillin, 100 µg/ml Streptomycin, 3 mM CaCl2, 0.5 mg/ml collagenase D, and 40 µg/ml DNaseI. After blocking in Fc-block, cells were stained against CD11c, MHCII, and CD45.2. The exact ratio of injected cells was quantified by analyzing the injection mixture prior to footpad injection. Where indicated DCs were simultaneously incubated with M14-23 antibody while stimulated with UPEC ON mutants.

## In vitro T cell assay

T cell assays were performed as described previously (*Leithner et al., 2021*). In brief, primary naïve CD4⁺ T cells were isolated from the spleen of OT-II mice (B6.Cg-Tg(TcraTcrb)425Cbn/J) using EasySep

Mouse CD4[+] T cell isolation kit (Stemcell Technologies) after homogenization with a 70 µm cell strainer and resuspending the cells in PBS supplemented with 2% FCS and 1 mM EDTA. T cells were co-cultured with DCs matured with CFT073 locked-ON (KT179) or OFF (KT180) at a ratio of 5:1 ($5 \times 10^4$ T cells:$1 \times 10^4$ DCs) in 96-well round bottom well plates in R10 medium.

### T cell activation

After 24 hr co-culture in the presence of 0.1 µg/ml OVA, medium was removed by spinning. Cells were incubated with Fc receptor block in FACS buffer and stained with anti-CD4, anti-CD69, and anti-CD62-L antibodies for 15 min at 4°C. After resuspending cells in FACS buffer, 100 µl were recorded on FACS Canto II (BD) and the ratio of CD69 to CD62L expression of CD4[+] T cells was analyzed by FlowJo software.

### T cell priming

T cells were stained with 5 µM CFSE stain in 5% FCS in PBS for 5 min at RT. After 30 min recovery in R10 medium at 37°C, cells were routinely checked for fluorescence.

After co-culturing with DCs for 4 days in the presence of 0.1 µg/ml OVA, medium was removed by spinning. Cells were stained with anti-CD4 antibody for 10 min at 4°C. After resuspending cells in FACS buffer and 7AAD viability stain, 100 µl were recorded on FACS Canto II (BD) (WT data) or on Cytoflex LX (BC) (*Itgb2*[−/−] data). The amount of T cells in proliferation was analyzed by FlowJo software.

### In vivo T cell assay

T cell assays were performed as described previously (*Leithner et al., 2021*). In brief, primary naïve CD4[+] T cells were isolated from the spleen of OT-II mice using EasySep Mouse CD4[+] T cell isolation kit after homogenization with a 70-µm cell strainer and resuspending the cells in PBS supplemented with 2% FCS and 1 mM EDTA. T cells were stained with 5 µM CFSE stain in 5% FCS in PBS for 5 min at RT. After 30 min recovery in R10 medium at 37°C, cells were routinely checked for fluorescence. $1 \times 10^6$ CFSE labeled CD4[+] T-cells were injected retro-orbital into 4- to 6-week-old Pep Boy (B6 CD45.1, B6.SJL-*Ptprc*[a] *Pepc*[b]/BoyJ) mice. 24 hr later $2.5 \times 10^5$ CFT073 locked-ON (KT179) or OFF (KT180) stimulated DCs, pre-loaded with 0.1 µg/ml OVA for 1 hr, in 25 µl PBS were injected subcutaneously into the hind footpad. 72 hr after injecting the T cells, mice were sacrificed and the popliteal lymph nodes were collected. Lymph nodes were ripped open and digested for 30 min at 37°C in complete DMEM supplemented with 2% FCS, 100 U/ml penicillin, 100 µg/ml streptomycin, 3 mM $CaCl_2$, 0.5 mg/ml collagenase D, and 40 µg/ml DNAseI. After blocking in Fc block, cells were stained against CD4 and CD45.2 for 10 min at 4°C. After resuspending cells in FACS buffer and 7AAD viability stain, the amount of T cells in proliferation was analyzed by FlowJo software.

### DC–T cell interaction time

Interaction time of DCs and T cells was measured as described previously (*Leithner et al., 2021*). In brief, glass bottom dishes were plasma cleaned for 2 min and coated with 1× poly-L-lysine in water for 10 min at RT. Dishes were washed two times with water and dried overnight. $1.5 \times 10^5$ DCs, pre-loaded with 0.1 µg/ml OVA for 1 hr 30 min, were mixed with $3 \times 10^5$ T cells and loaded onto the coated dishes in a total volume of 300 µl. Images were taken every 30 s for a total of 6 hr on bright field microscope using ×20 magnification and an exposure of 20ms. Images were processed with Fiji.

### Quantification and statistical analysis

Data are represented as means ± standard deviations. Statistics were performed using GraphPad Prism version 9.0.2 for Windows. Statistical details for each experiment can be found in the respective figure legends. Significance was defined as follows: *$p < 0.1$, **$p < 0.05$, ***$p < 0.01$, ****$p < 0.001$.

## Acknowledgements

We thank Ulrich Dobrindt for providing UPEC strains CFT073, UTI89, and 536, Frank Assen, Vlad Gavra, Maximilian Götz, Bor Kavčič, Jonna Alanko, and Eva Kiermaier for help with experiments and Robert Hauschild, Julian Stopp, and Saren Tasciyan for help with data analysis. We thank the IST Austria Scientific Service Units, especially the Bioimaging facility, the Preclinical facility and the

Electron microscopy facility for technical support, Jakob Wallner and all members of the Guet and Sixt lab for fruitful discussions and Daria Siekhaus for critically reading the manuscript. This work was supported by grants from the Austrian Research Promotion Agency (FEMtech 868984) to IG, the European Research Council (CoG 724373), and the Austrian Science Fund (FWF P29911) to MS.

## Additional information

### Competing interests

Kathrin Tomasek, Calin C Guet, Michael Sixt: is an inventor on patent application 21170193.3 ("Methods determining the potential of drug for treating bacterial infections and composition for treating bacterial infections"). Michael S Lukesch: is an inventor on patent application 21170193.3 ("Methods determining the potential of drug for treating bacterial infections and composition for treating bacterial infections"), and is affiliated with VALANX Biotech GmbH. The author has no financial interests to declare. The other authors declare that no competing interests exist.

### Funding

| Funder | Grant reference number | Author |
|---|---|---|
| European Research Council | CoG 724373 | Michael Sixt |
| Austrian Science Fund | FWF P29911 | Michael Sixt |
| Österreichische Forschungsförderungsgesellschaft | FEMtech 868984 | Ivana Glatzova |

The funders had no role in study design, data collection, and interpretation, or the decision to submit the work for publication.

### Author contributions

Kathrin Tomasek, Conceptualization, Data curation, Formal analysis, Investigation, Methodology, Writing – original draft, Writing – review and editing, Visualization, Validation, Supervision; Alexander Leithner, Data curation, Investigation, Writing – review and editing, Conceptualization; Ivana Glatzova, Investigation, Writing – review and editing, Data curation; Michael S Lukesch, Investigation, Writing – review and editing, Conceptualization; Calin C Guet, Conceptualization, Methodology, Project administration, Writing – review and editing, Funding acquisition, Supervision; Michael Sixt, Conceptualization, Funding acquisition, Methodology, Project administration, Writing – review and editing, Supervision

### Author ORCIDs

Kathrin Tomasek http://orcid.org/0000-0003-3768-877X
Alexander Leithner http://orcid.org/0000-0002-1073-744X
Calin C Guet http://orcid.org/0000-0001-6220-2052
Michael Sixt http://orcid.org/0000-0002-6620-9179

### Ethics

All animal experiments are in accordance with the Austrian law for animal experiments. Permission was granted by the Austrian Federal Ministry of Science, Research and Economy (identification code: BMWFW 66.018/0010-WF/V/3b/2016). Mice were bred and maintained at the local animal facility in accordance IST Austria Ethical Committee or purchased from Charles River and maintained at the local animal facility in accordance with IST Austria Ethical Committee.

### Decision letter and Author response

Decision letter https://doi.org/10.7554/eLife.78995.sa1
Author response https://doi.org/10.7554/eLife.78995.sa2

# Additional files

## Supplementary files
- Supplementary file 1. PDB file of mouse CD14 and FimH.
- Supplementary file 2. PDB file of human CD14 and FimH.
- Supplementary file 3. Predicted free binding energies of mouse CD14, TLR4, and CD48 to FimH.
- Supplementary file 4. Amino acids of FimH responsible for binding mouse CD14 and mannose, and amino acids of mouse CD14 responsible for FimH and lipopolysaccharide (LPS) binding.
- Supplementary file 5. Predicted free binding energy of human CD14 and FimH.
- Transparent reporting form
- Source code 1. Pre-processing of files from the in vitro 3D collagen migration images. Removes edge effects and stabilizes image. Resulting file is used for further analysis in migrationspeedREP script.
- Source code 2. Returns migration speed of single cells in the in vitro 2D migration assay of adherent cells.
- Source code 3. Reads data from the Tracking_pre-processing_for_brightfield script. Returns average of migration speed and cell displacement in the in vitro 3D collagen migration assay as pixels/min.
- Source code 4. Pre-processing of the ex vivo migration images. Manual outlining of lymph vessels and adjustment of fluorescent intensities of dendritic cell (DC) signals for further analysis in LVmeanDCarea script.
- Source code 5. Reads data from the lymphatics script. Returns average of fluorescence intensity of dendritic cells (DCs) inside and outside of lymph vessels.
- Source code 6. Plots maximum projection images of the ex vivo migration images.

## Data availability
All data are included in the manuscript. Source data are uploaded with this manuscript.

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
