## [Editor Report]

Sixt and colleagues demonstrate that a protein that sits at the tip of the pilus of a gram-negative bacterium exerts immune evasion functions by binding CD14, the co-receptor of TLR4. The identification of this pathway may help to develop new therapeutic approaches to intervene against recurrent infections and/or inflammatory diseases of the gut.

---

## [Decision Letter]

**Decision letter after peer review:**

[Editors’ note: the authors submitted for reconsideration following the decision after peer review. What follows is the decision letter after the first round of review.]

Thank you for submitting the paper "Type 1 piliated uropathogenic *Escherichia coli* hijack the host immune response by binding to CD14" for consideration by *eLife*. Your article has been reviewed by 3 peer reviewers, one of whom is a member of our Board of Reviewing Editors, and the evaluation has been overseen by a Senior Editor. The reviewers have opted to remain anonymous.

Comments to the Authors:

We are sorry to say that, after consultation with the reviewers, we have decided that this work in its present form will not be considered further for publication by *eLife*.

Specifically, all Reviewers feel the topic is of great interest but that more mechanistic insights are necessary and that relevance of the findings reported should be also assessed in vivo, at least in the context of DC migration. We anticipate that addressing these comments would take more than the 2-month time we allow for revisions, we are therefore rejecting the manuscript at this point. However, should the authors be able to address the points mentioned above, we would be delighted to receive a revised manuscript.

*Reviewer #1:*

Initially the authors described the capacity of pilus-expressing *E. coli* to reduce T cell proliferation, co-stimulatory molecule induction on DCs, and to increase DC-T cell interaction time (via CD11b-activation) in vitro. Also, that the migratory capacity into lymphatics of DCs exposed to pilus+ *E. coli* was significantly decreased.

The authors showed that reduced co-stimulatory molecule induction and potentiated integrin activation are events independently regulated, but how this regulation takes place remains completely overlooked. Also, if production of inflammatory mediators -such as pro-inflammatory cytokines or interferons- is also affected remains unknown.

Next, the authors focused on the mechanism that allows the pilus to change the functions of DCs. Because of previous literature, authors tested the involvement of TLR4 and found that in Tlr4-deficient cells the presence of pili still affects adhesion and migration into the lymphatics. The authors concluded that TLR4 is not involved in these processes. An important information that lacks here is, though, the relevance of TLR4 in regulating co-stimulatory molecules and inflammatory cytokines, as well as T cell proliferation.

Next the authors focused on CD14 for its links with integrin activity and cell adhesion. The authors convincingly show that CD14 engagement decreases the migratory capacity of DCs and dampen co-stimulatory molecules induction by pilus+ *E. coli* (although also here any info about T cells is missing). Most importantly, the authors demonstrated that the presence of FimH expressed by pathogenic *E. coli* is necessary for these functions. Notably, expression of pathogenic FimH on pili of non-pathogenic coli did not reproduced the phenotype. These data raise important questions about the minimal requirements necessary for FimH to exert its functions.

Finally, the authors identified 3 aminoacidic residues of FimH necessary to bind CD14 and either mutated them or inhibited their binding capacity with D-mannose, a D-mannose derivative, as well as an antibody directed against CD14. These experiments showed that at least co-stimulatory molecule expression was partially rescued by these treatments. Based on these data, the authors propose possible future therapeutic interventions aimed at targeting FimH and CD14 interaction.

Overall, this is a well-designed, but still quite preliminary, work that lacks many mechanistic insights and a proof that the interaction between CD14 and FimH really plays a relevant role in vivo during an infection.

In my opinion, there are two major points that need to be addressed to make this paper a strong candidate for *eLife*: (i) more mechanistic insights on how CD14 engagement by FimH modulate DC activation; (ii) in vivo relevance of the pathways identified.

Regarding the mechanistic insights:

How the global transcriptome of DCs exposed to pilus-positive or negative bacteria is affected? Are interferons or other pro-inflammatory cytokines differentially regulated? The authors solely showed that TLR4-deficency does not affect adhesion and migration into lymphatics, but not that DCs respond in the same manner to pilus+ or pilus-negative bacteria.

How the engagement of CD14 by FimH modulates the activation of integrins or the inhibition of co-stimulatory molecules? Is FimH preventing binding of LPS by CD14 and, thus, changing the signaling capacity of TLR4?

CD14 is internalized upon LPS binding to drive IFN-I production. Does this happen with FimH^+^ bacteria? CD14 has also been shown to exert TLR4-independent funcitons such as calcium influxes and NFAT activation. Is any of those potentiated or prevented by the encounter of FimH^+^ bacteria?

How does this influence either co-stimulatory molecule expression? And what about IFN-I production? The authors have all the tools and capacity to answer these questions.

Regarding the in vivo experiments:

Relevance of the pathways and mechanisms described in the paper needs to be proved in a mouse model of UPEC infection.

Other points:

Please add the analysis on T cell proliferation as well as the other analyses missing as reported in the Review.

Authors should further dissect the phenotype of T cells immobilized on DCs in vitro. They claim this reduces activation, but this experiment may also suggest the opposite. Activation of OT-II cells using CD11b-deficient DCs should be performed.

*Reviewer #2:*

In this study, Tomasek et al. tested the hypothesis that type 1 pili, a pre-adapted virulence factor used by uropathogenic *E. coli* (UPEC) to interact with host infected cells, alter dendritic cell (DC) activation. They found that type 1 pili interfere with CD14 activities, leading to defective DC migration, costimulatory molecule expression and antigen presentation. The main strength of the study is the dissection of the structural interplay between FimH and CD14, as it convincingly shows that the two molecules interact. On the other hand, analysis of the functional impact of type 1 pili on DC activation as well as on CD14 activities could be improved. Overall, this study provides a conceptual advance in our understanding of host-pathogen interactions and highlight CD14 as a possible target for the treatment of persistent bacterial infections.

Most of the work is based on the analysis of DC responses to bacteria with (ON) or without (OFF) type 1 pili. Incubation of DCs with UPEC ON led to reduced antigen (OVA) presentation and T cell proliferation; lower induction of costimulatory molecules by DCs; higher integrin (Cd11b) activation; tighter adhesion of DCs to serum-coated surfaces; reduced DC migration ex vivo and in explanted tissues. CD14ko, but not TLR4ko, DCs show a normalized migration behavior and costimulatory molecule expression in response to UPEC ON.

The authors conclude that type 1 pili limit Ag presentation capacities of DCs by interfering with costimulatory molecule expression and by causing integrin hyperactivation, leading to immobilization on ECM and defective migration. They also propose that this occurs in a CD14-dependent manner.

- The experiments in support of a causative role of Cd11b hyperactivation in defective DC migration are not entrely convincing, as they only show that UPEC ON stimulation does not immobilize DCs on type 1 collagen (which is not bound by Cd11b). To sustain their point, the authors may test whether Cd11b hyperactivation by means other than UPEC ON exposure also results in DC immobilization, or test whether blockade of CD11b hyperactivation (or of Cd11b) restore migration patterns of DCs stimulated with UPEC ON.

- Whether UPEC ON-dependent DC immobilization has any consequence in vivo for antigen presentation also remains unclear. The authors could test whether UPEC ON or OFF are differentially capable of eliciting DC recruitment and interaction with T cells to lymph nodes.

- The authors find that CD14ko are not sensitive to UPEC ON-mediated alterations of migration, and they conclude that CD14 is required for increased integrin activity. This conclusion is not fully supported, in my view. It would be important to link CD14 and Cd11b hyperactivation using an independent, and possibly more direct, readout of integrin hyperactivation. More generally, a detailed characterization of how CD14ko DC respond to UPEC ON or UPEC OFF would be helpful to support the proposed link between the type 1 pili and CD14.

- It remains unclear whether interference with CD14 activity by UPEC ON has any impact in vivo. The study would be much improved if the authors could compare host survival and bacterial persistence upon infection of wt (and possibly CD14ko) mice with UPEC On and UPEC OFF.

The authors identify a potential binding site between FimH and CD14 and then convincingly show that these molecules physically interact. FimH therefore appears to be a critical determinant of the pathogenic effect of type 1 pili in DCs.

- The interaction site between FimH and CD14 is said to occur "in an area not involved in LPS binding". It would be important if the authors supported this claim experimentally, by measuring whether FimH alters inflammatory gene expression and/or cytokine expression and in LPS-stimulated DCs. More generally, the study would greatly benefit it the authors could

- CD14 acts in DCs in a ligand dose-dependent manner. Can the authors show DC responses to different concentration of UPEC ON/OFF?

*Reviewer #3:*

The authors explore the ability of the pili, a virulence factor typical of *Enterobacteriaceae* and of particular relevance in the study of uropathogenic *E. coli* (UPEC), to regulate the host immune response. Pili are particularly interesting as their expression can be turned on and off cyclically by bacteria due to phase variation, which can allow modulation of the virulence of isogenic bacteria. While it is known that pili modulate bacterial adhesion to host cells, little is known of the direct immunomodulatory properties of pili. Here the authors use a clever mutant *E. coli* model which is locked either in the ON (pilus expressor) or OFF( pilus non expressor) condition to study the ability of the pilus to influence dendritic cell (DCs) responses. The authors convincingly demonstrate that: (a) Pilus expressing (ON)-*E. coli* can inhibit the capacity of DCs to stimulate transgenic T cells in vitro, (b) ON-*E. coli* can down-modulate the ability of DCs to upregulate the costimulatory molecule CD40 and to a lesser extent CD80 and CD86, (c) ON-*E. coli* reduces DCs motility by enhancing the activation of beta2 integrins, which results in DCs being retained in the tissues and migrating less efficiently into the lymph, (d) CD14 is the receptor for pili proteins and is interaction between pili and CD14 is responsible for modulating some DCs responses.

While the experiments are convincing and well controlled, the authors fail to evaluate key innate functions of dendritic cells, such as cytokine production, where CD14 plays an important regulatory role, either indirectly by regulating TLR internalization and activation of the endosomal pathway, or directly by activating calcium signaling and NFAT activation. It is also unclear whether CD14 binding is blocking its ability to respond to other ligands (i.e. LPS). This tampers the impact of the conclusions. Moreover the absence of any in vivo infection experiment, tampers the enthusiasm for the paper. It's difficult to interpret the claim that pili influence DCs ability to activate T cells with only limited in vitro experiments. DCs ability to influence adaptive immunity is dependent on several things, including antigen transport, cytokine production, cell-cell interaction, migration etc…that cannot be fully recapitulated in vitro. Also, DCs exert important innate immune functions locally in the tissues, that are not only the activation of T cells and do not require their migration into the lymph, an aspect that is not taken in consideration by the authors.In conclusion, while the experiments are sound and controlled, and the idea is of great general interest in the context of host pathogen interactions, the analyses are insufficient to understand the immunomodulatory properties of pili with a level of depth that can be meaningful in an infection context.

As mentioned above, while the paper does a good job at identifying pili as structures able to influence DC responses through CD14, they fail to evaluate a number of important functions of dendritic cells that are directly regulated by the receptor CD14, such as cytokine production downstream of TLR4 and/or directly downstream to CD14. Moreover, they do not go into sufficient depth neither regarding the molecular mechanism downstream to CD14 that could explain the observed phenotypes (are pili stimulating responses downstream of CD14? Are they inhibiting LPS binding? Are they inhibiting interactions between CD14 and TLR4 or TLR2?) , nor the ability of pili to influence the overall outcome and/or the activation of DCs during an infection in vivo.

The authors also discuss their results stressing that pili inhibit the capacity of DCs to activate T cells, however, only one initial experiment evaluating T cell activation is performed, in an in vitro setting with transgenic TCR bearing T cells. While the authors demonstrate that the time of contact between T cells and DCs is augmented in the presence of pilus expressing bacteria, they do not explore the consequences of this enhanced contact and the relative impact of integrin activation vs costimulatory molecules expression in influencing T cell responses. While the topic is of great potential interest and the authors have devised an interesting tool to specifically study the importance of pili, they need to add depth to their work in order to take complete advantage of the tool they created and to help to answer the question they have posed.

Even taking into account the difficulty and the regulatory burden of animal experimentation, the authors could do a better job in dissecting the molecular mechanisms underlying the ability of pili to inhibit DCs and the dynamics of DCs-T cell interaction.

[Editors’ note: further revisions were suggested prior to acceptance, as described below.]

Thank you for resubmitting your work entitled "Type 1 piliated uropathogenic *Escherichia coli* hijack the host immune response by binding to CD14" for further consideration by *eLife*. Your revised article has been evaluated by Carla Rothlin (Senior Editor) and a Reviewing Editor.

The manuscript has been improved but there are some remaining issues that need to be addressed, as outlined below:

i) Incorporate the changes to the text requested by Reviewer 1.

ii) Concerning the requests of Reviewer 2, the authors should present a side-by-side comparison of cytokines produced by WT DCs, CD14 ko DCs and FK506-treated WT DCs to formally prove a reduction in NFAT-dependent cytokines by CD14 KO DCs and FK506-treated DCs in response to "ON" bacteria.

*Reviewer #1:*

The authors incorporated many requests previously made. In particular, they added new exciting results re NFAT relevance in their model and about cell-intrinsic and cell-extrinsic requirements. in vivo data, though only based on in vitro stimulated DCs, nicely complement the findings of the authors. The authors also analyzed additional isolates that suggest bacterium-specific behaviors.

Although the authors did not perform in vivo infections, and some of their (new) findings open more mechanistic questions, the elegant ON/OFF model utilized, the identification of CD14 as a key regulator of responses to FimH, together with the new data provided make this new version of the manuscript exciting.

I have only a few changes to request:

1) Please properly cite in line 248, 249 the citation reported in line 1629.

2) Re Figure 3G: Please, stress that RANTES is an ISG and, thus, the capacity of CD14 to drive interferon-dependent responses upon LPS encounter is not altered by the presence of FimH. Ideally, authors should have measured type I interferons to prove that CD14-intrinsic responses to LPS are not altered, but this point can be just mentioned, also based on data on CD14 internalization.

3) Always re Figure 3G: I would strongly encourage the authors to remove IFNγ from the cytokines measured. Murine DCs do not efficiently produce IFNγ in response to LPS/gram-negative bacteria. And indeed the authors measured extremely low levels, if any, of this cytokine.

4) Please add to the discussion or at the end of the paragraph a brief line about the results shown in Figure 3I about the maintenance of increased integrin activity in CD14-/- cells exposed to ON bacteria.

*Reviewer #2:*

The authors have performed a number of new experiments to address my initial concerns. The new manuscript shows that Pili act through the CD14-NFAT axis to induce the production of immunomodulatory cytokines that, in turn decrease upregulation of costimulatory molecules. The authors also added in vivo data modeling DCs migration by injecting OVA loaded-, *E. coli*-activated-, DCs in the footpad and analyzing migration and proliferation that demonstrate that the combination of defects in migration as well as costimulatory molecule affects T cell proliferation, albeit with a small size effect, and that the effect is not cell autonomous (consistently with the involvement of NFAT mediated cytokine production in the phenotype).

The new data support idea that CD14 control of cytokine production through NFAT can induce the downregulation of costimulatory molecules, however I believe that some important controls should be included:

The authors do not show any comparison between CD14 Ko DCs and WT DCs in term of cytokine production, and do not show whether CD14 KO and FK506 have the same effect on cytokine production and downregulation of costimulatory molecules.

While it might be beyond the authors scope, it would be interesting to identify which NFAT dependent cytokine is mediating the phenotype observed.

Moreover the small size of the in vivo effect, as well as the fact that it is not cell autonomous somewhat lower the enthusiasm for the conclusions, in particular, due to the fact that pili expression can be turned on and off in vivo, thus mimicking the effect of a mixed stimulation. Importantly assessment of mixed transfer on T cell proliferation is missing!

Some other concerns include:

a) WT, CD14 and TLR4 KO cells are never depicted in the same graph and therefore I deduce that the experiments on each cell genotype are independent experiments. These comparisons should be done in the same experiment to be able to compare KO with WT. Indeed the author state in line: 232-234 " It was previously shown that tlr4-/- DCs express reduced levels of co-stimulatory molecules (Shen et al., 2008) and this was also true for our in vitro generated tlr4-/- DCs". However, TLR4 -/- DCs are never compared to WT DCs in the manuscript.

b) While internalization of CD14 Is assessed, production of type I interferons is never assessed and should be considered as CD14 trafficking of TLR4 can influence IFN production.

c) While CD80 and CD86 expression are clearly depending on CD14, CD40 expression seems to be TLR4 dependent. The author should discuss this point.

d) In line 244 LPS driven cytokines is incorrect. TLR4 driven or NfKB dependent cytokines would be more rigorous.

Overall, while the manuscript has significantly improved, some pieces of the puzzle are still missing and the modest in vivo effect, in an artificial model of DCs migration and function, leaves open the question of the relevance of this mechanism during infection.

*Reviewer #3 (Recommendations for the authors):*

The authors have performed a large set of experiments that satisfactorily address most of my previous criticisms – especially for what concerns the functional impact of type 1 pili on CD14-mediated DC activation. In my view, the revised manuscript is a strong candidate for publication in *eLife*.

---

## [Author Response]

[Editors’ note: the authors resubmitted a revised version of the paper for consideration. What follows is the authors’ response to the first round of review.]

Reviewer #1:Initially the authors described the capacity of pilus-expressing *E. coli* to reduce T cell proliferation, co-stimulatory molecule induction on DCs, and to increase DC-T cell interaction time (via CD11b-activation) in vitro. Also, that the migratory capacity into lymphatics of DCs exposed to pilus+ *E. coli* was significantly decreased.The authors showed that reduced co-stimulatory molecule induction and potentiated integrin activation are events independently regulated, but how this regulation takes place remains completely overlooked. Also, if production of inflammatory mediators -such as pro-inflammatory cytokines or interferons- is also affected remains unknown.

We thank reviewer 1 for the comments on our manuscript. We performed additional experiments to gain deeper insight into the signaling pathway downstream of the FimH-CD14 interaction. We analyzed cytokine levels in the supernatant of UPEC ON and OFF stimulated DCs (line 244-247 and line 321-325, Figure 3G und 4H). We found that ON stimulation led to increased expression of several immune-modulatory cytokines, mainly linked to NFAT pathway activation (eg IL-4, IL-10, GM-CSF), whereas cytokines related to activation of DCs via LPS (eg IL-6 and TNFalpha) were equally expressed upon ON and OFF stimulation.

Next, the authors focused on the mechanism that allows the pilus to change the functions of DCs. Because of previous literature, authors tested the involvement of TLR4 and found that in Tlr4-deficient cells the presence of pili still affects adhesion and migration into the lymphatics. The authors concluded that TLR4 is not involved in these processes. An important information that lacks here is, though, the relevance of TLR4 in regulating co-stimulatory molecules and inflammatory cytokines, as well as T cell proliferation.

We tested the expression of co-stimulatory molecules on tlr4 -/- DCs (line 231-236, Figure 3C and Supplementary Figure 3C). In line with the increase in adhesion and the decrease in migration, tlr4 null DCs displayed decreased expression of co-stimulatory molecules after UPEC ON stimulation, as compared to UPEC OFF stimulation.

Next the authors focused on CD14 for its links with integrin activity and cell adhesion. The authors convincingly show that CD14 engagement decreases the migratory capacity of DCs and dampen co-stimulatory molecules induction by pilus+ *E. coli* (although also here any info about T cells is missing). Most importantly, the authors demonstrated that the presence of FimH expressed by pathogenic *E. coli* is necessary for these functions. Notably, expression of pathogenic FimH on pili of non-pathogenic coli did not reproduced the phenotype. These data raise important questions about the minimal requirements necessary for FimH to exert its functions.

We agree with the reviewer that although FimH is necessary for the observed immune-modulatory phenotypes, we did not find (pathogenic) FimH to be sufficient. To address this point, we performed adhesion assays (as an indirect measure also for the migration potential) and expression of costimulatory molecules using ON and OFF mutants of two additional clinical isolates (strains UTI89 and 536) to get an understanding if this is a UPEC-wide phenomenon (line 310-315, Supplementary Figure 6E-F). Interestingly, the strength of the immune-modulatory effects varied in the different strains – some affect adhesion, whereas others do not. Only CFT073 affected expression of all co-stimulatory molecules. We conclude that CFT073 expresses virulence factors that act downstream of the FimHdependent CD14 signaling and thereby modulate the observed phenotypes.

Finally, the authors identified 3 aminoacidic residues of FimH necessary to bind CD14 and either mutated them or inhibited their binding capacity with D-mannose, a D-mannose derivative, as well as an antibody directed against CD14. These experiments showed that at least co-stimulatory molecule expression was partially rescued by these treatments. Based on these data, the authors propose possible future therapeutic interventions aimed at targeting FimH and CD14 interaction.Overall, this is a well-designed, but still quite preliminary, work that lacks many mechanistic insights and a proof that the interaction between CD14 and FimH really plays a relevant role in vivo during an infection.

We performed a series of experiments (both ex vivo and in vivo) to address the mechanistic aspects of the FimH/CD14 interaction. We added a new figure to support these findings (Figure 7).

First, we identified the NFAT pathway to be overactivated downstream of FimH binding to CD14 (line 316-321, Figure 4G and Supplementary Figure 4D) and that this leads to increased expression of NFATdriven cytokines (line 321-325, Figure 4H).

Second, to verify that the interaction between CD14 and FimH plays a relevant role in vivo during an infection, we performed in vivo migration assays using UPEC ON and OFF stimulated DCs, and additionally blocked CD14 during stimulation of DCs using the blocking CD14 antibody M14-23 (line 396-402, Figure 7C-D). Additionally, we analyzed in vivo T cell proliferation after UPEC ON and OFF stimulation (line 381-388, Figure 7A). We found migration of DCs from the footpad into the popliteal lymph node to be reduced after ON stimulation, compared to OFF stimulation, although due to the spread of the data in vivo this effect was not significant (Figure 7C), whereas treating DCs with the blocking CD14 antibody restored in vivo migration of ON stimulated DCs fully (Figure 7D). Interestingly, as soon as we mixed ON and OFF stimulated DCs before injection (Figure 7F) or performed ex vivo ear crawl in migration assays with mixed cells (Figure 7A), we found that the same amount of DCs reached the lymph node, irrespective of the stimulus. We assume that these effects are due to non-cell autonomous effects the DCs face once they are in close contact with differentially stimulated DCs (Bardou et al., EMBOJ, 2021). Interestingly, although the overall amount of in vivo divided T cells was the same between ON and OFF stimulation, a greater fraction of T cells reached further rounds of division after OFF stimulation, compared to after ON stimulation – reflected in the proliferation index (Figure 7A). We hypothesize that ON stimulation therefore also affects in vivo T cell priming, but given that inside the host antigen transfer to resident DCs happens (Kleindienst and Brocker, The Journal of Immunity, 2003), we think that the effect is not as pronounced as in vitro. We therefore think that the immune-modulatory effect we found is of special interest when looking at cell-cell interactions.

In my opinion, there are two major points that need to be addressed to make this paper a strong candidate for eLife: (i) more mechanistic insights on how CD14 engagement by FimH modulate DC activation; (ii) in vivo relevance of the pathways identified.

We now performed additional experiments to answer those questions (see outlined below).

Regarding the mechanistic insights:How the global transcriptome of DCs exposed to pilus-positive or negative bacteria is affected? Are interferons or other pro-inflammatory cytokines differentially regulated? The authors solely showed that TLR4-deficency does not affect adhesion and migration into lymphatics, but not that DCs respond in the same manner to pilus+ or pilus-negative bacteria.How the engagement of CD14 by FimH modulates the activation of integrins or the inhibition of co-stimulatory molecules? Is FimH preventing binding of LPS by CD14 and, thus, changing the signaling capacity of TLR4?CD14 is internalized upon LPS binding to drive IFN-I production. Does this happen with FimH^+^ bacteria? CD14 has also been shown to exert TLR4-independent funcitons such as calcium influxes and NFAT activation. Is any of those potentiated or prevented by the encounter of FimH^+^ bacteria?How does this influence either co-stimulatory molecule expression? And what about IFN-I production? The authors have all the tools and capacity to answer these questions.

We now detected cytokine levels after UPEC ON and OFF stimulation and we found LPS-driven cytokines to be similarly expressed between ON and OFF stimulated DCs (line 244-247, Figure 3G) and only minimal differences in MHCII levels (Supplementary Figure 1D). This indicates that DCs do react to UPEC ON mutants in an LPS dependent manner. Additionally, we tried to override the negative effects of UPEC ON mutants on DC adhesion by adding excessive amounts of exogenous LPS (200ng/ml). Overstimulation with LPS did not lift the effect ON mutants had on adhesion of DCs (data not added to the manuscript), demonstrating that the FimH-CD14 interaction is actin in a dominant manner.

The molecular details, how the FimH CD14 binding affects downstream signaling, such as the NFAT pathway (line 316-321, Figure 4G and Supplementary Figure 4D) are unclear, given that CD14 is a GPI anchored protein and thus has to act via co-receptors. We found equal rates of CD14 internalized upon UPEC ON and OFF stimulation (line 247-249, Figure 3H) and based on literature (Zanoni 2009) we hypothesize that activation of the NFAT pathway involves Ca signaling, leading to overactivation of the NFAT pathway and thus increased production of NFAT-driven cytokines (line 321-325, Figure 4H). Accordingly, we found that blocking activation of calmodulin (using FK506) prior to ON stimulation, rescued the expression of co-stimulatory molecules (line 316-321, Figure 4G and Supplementary Figure 4D).

We assayed expression of co-stimulatory molecules on tlr4 -/- DCs and found expression of costimulatory molecules still reduced after ON stimulation (line 231-236, Figure 3C and Supplementary Figure 3C) again suggesting that CD14 signaling is the rate-limiting pathway.

We also performed in vitro T cell proliferation assays using beta2 -/- DCs to dissect the effects of increased adhesion and decreased co-stimulatory molecules on T cell priming (line 211-216, Figure 2G und Supplementary Figure 2G). Beta2 integrin -/- DCs had a lower baseline-efficiency in supporting T cell proliferation when compared to WT DCs. Importantly, proliferation was still decreased after ON stimulation compared to OFF stimulation of beta2 -/- DCs. We conclude, that not hyperactivation of integrins but decreased co-stimulation is the dominant cause of reduced T cell priming. Additionally, cd14 -/- DCs still show increased levels of integrin activity after ON stimulation (line 249-251, Figure 3I and Supplementary Figure 3G).

Finally, we performed in vivo migration and T cell proliferation assays (Iine 381-412, Figure 7) and we found immunosuppression of DCs by ON UPECs to be modulated by non-cell autonomous effects.

Regarding the in vivo experiments:Relevance of the pathways and mechanisms described in the paper needs to be proved in a mouse model of UPEC infection.

Although we agree that it will be very interesting to see how the molecular interaction we describe affects urinary tract infections (UTIs), we think that this is beyond the scope of this manuscript. UTIs are extremely complex pathogenetic sequences, and type 1 pili and FimH have pleiotropic roles in this process, including invasion into epithelia. Here, the fact that the FimH-CD14 interaction site overlaps with the mannose binding site that mediates invasion is especially challenging. The immune responses following invasion involve both innate and adaptive responses, which are difficult to dissect. As a natural infection always occurs with bacteria that can phase-vary between ON and OFF, it will be necessary to first characterize if and how UTIs proceed after infection with the pure ON and OFF mutants. In short, we argue that our findings represent the groundwork that will hopefully be picked up by more clinically oriented researchers who have proper animal models and in vivo expertise at hand to dissect how the pathway we identified can be utilized to target UTIs.

Other points:Please add the analysis on T cell proliferation as well as the other analyses missing as reported in the Review.Authors should further dissect the phenotype of T cells immobilized on DCs in vitro. They claim this reduces activation, but this experiment may also suggest the opposite. Activation of OT-II cells using CD11b-deficient DCs should be performed.

We now performed T cell assays in vitro using beta2 -/- DCs (line 211-216, Figure 2G). We found activation of OT-II T cells still reduced after contact with ON stimulated DCs, compared to OFF stimulation. This proves that beta2 knockout DCs are unable to activate T cells after ON stimulation, giving the lack of co-stimulatory molecules.

Reviewer #2:In this study, Tomasek et al. tested the hypothesis that type 1 pili, a pre-adapted virulence factor used by uropathogenic *E. coli* (UPEC) to interact with host infected cells, alter dendritic cell (DC) activation. They found that type 1 pili interfere with CD14 activities, leading to defective DC migration, costimulatory molecule expression and antigen presentation. The main strength of the study is the dissection of the structural interplay between FimH and CD14, as it convincingly shows that the two molecules interact. On the other hand, analysis of the functional impact of type 1 pili on DC activation as well as on CD14 activities could be improved. Overall, this study provides a conceptual advance in our understanding of host-pathogen interactions and highlight CD14 as a possible target for the treatment of persistent bacterial infections.

We thank reviewer 2 for the encouraging words on our work as well as for pointing out the limitations. To address these, we performed a new set of experiments that tested the functional impact of type 1 pili on DCs by identifying the NFAT pathway as one pathway upregulated upon UPEC ON stimulation (line 316-321, Figure 4G and Supplementary Figure 4D) and we analyzed cytokine expression of UPEC ON and OFF stimulated DCs (line 321-325, Figure 4H). As also mentioned in the reply to reviewer 1, we found that ON stimulation leads to overactivation of the NFAT pathway, most likely by increased calcium influx as already known upon CD14 activation (Zanoni 2009), and this lead to increased expression of several cytokines such as IL-4, IL-10 and GM-CSF. Additionally, we found LPS-driven cytokines, such as IL-6, TNF-α and IL-1beta, equally expressed after ON and OFF stimulation (line 244-247, Figure 3G) as well internalization of CD14 receptor line 247-249, Figure 3H indicating binding of FimH to CD14 does not inhibit LPS-driven activation of DCs. Finally, we performed in vivo migration and T cell priming assays to show the in vivo relevance of our findings that we show in a new figure (Iine 381-412, Figure 7) and we observed a strong non-cell autonomous effect indicating the relevance of our findings on a cell-to-cell basis.

Most of the work is based on the analysis of DC responses to bacteria with (ON) or without (OFF) type 1 pili. Incubation of DCs with UPEC ON led to reduced antigen (OVA) presentation and T cell proliferation; lower induction of costimulatory molecules by DCs; higher integrin (Cd11b) activation; tighter adhesion of DCs to serum-coated surfaces; reduced DC migration ex vivo and in explanted tissues. CD14ko, but not TLR4ko, DCs show a normalized migration behavior and costimulatory molecule expression in response to UPEC ON.The authors conclude that type 1 pili limit Ag presentation capacities of DCs by interfering with costimulatory molecule expression and by causing integrin hyperactivation, leading to immobilization on ECM and defective migration. They also propose that this occurs in a CD14-dependent manner.- The experiments in support of a causative role of Cd11b hyperactivation in defective DC migration are not entrely convincing, as they only show that UPEC ON stimulation does not immobilize DCs on type 1 collagen (which is not bound by Cd11b). To sustain their point, the authors may test whether Cd11b hyperactivation by means other than UPEC ON exposure also results in DC immobilization, or test whether blockade of CD11b hyperactivation (or of Cd11b) restore migration patterns of DCs stimulated with UPEC ON.

We performed migration assays of adherent DCs after ON and OFF stimulation on 2D surfaces in the presence of the chemokine CCL19. We found migration of adherent DCs after ON stimulation almost absent and therefore conclude that CD11b overactivation by UPEC ON mutants render DCs immobile (line 183-185, Supplementary Figure 2C).

- Whether UPEC ON-dependent DC immobilization has any consequence in vivo for antigen presentation also remains unclear. The authors could test whether UPEC ON or OFF are differentially capable of eliciting DC recruitment and interaction with T cells to lymph nodes.

To verify that the interaction between CD14 and FimH plays a relevant role in vivo during an infection, we performed in vivo migration assays using UPEC ON and OFF stimulated DCs and we analyzed in vivo T cell proliferation after UPEC ON and OFF stimulation (line 381-412, Figure 7). We comment on the results extensively in the reply to reviewer 1. Briefly, we found migration of DCs from the footpad into the popliteal lymph node reduced after ON stimulation, compared to OFF stimulation. However, as soon as we mixed ON and OFF stimulated DCs before injection, we found the same amount of DCs reaching the lymph node irrespective of the stimulus. We assume that these effects are due to noncell autonomous effects the DCs face once the DCs are in close contact with differentially stimulated DCs (Bardou et al., EMBOJ, 2021). T cell priming in vivo did not seem different after ON and OFF stimulation when looking at the overall amount of dividing T cells, however less T cells reached further rounds of division after ON stimulation reflected in the proliferation index. We hypothesize that ON stimulation therefore also affects in vivo T cell priming, but given that inside the host, antigen transfer to resident DCs in the lymph node occurs (Kleindienst and Brocker, The Journal of Immunity, 2003), we think that the effect is not as pronounced as in vitro.

- The authors find that CD14ko are not sensitive to UPEC ON-mediated alterations of migration, and they conclude that CD14 is required for increased integrin activity. This conclusion is not fully supported, in my view. It would be important to link CD14 and Cd11b hyperactivation using an independent, and possibly more direct, readout of integrin hyperactivation. More generally, a detailed characterization of how CD14ko DC respond to UPEC ON or UPEC OFF would be helpful to support the proposed link between the type 1 pili and CD14.

The most direct readout we can come up with to assay integrin activation is the integrin activity measurement in flow cytometry, as it reflects best what is actually happening on the integrin level itself without the need of an indirect readout such as migration. We therefore stained for active and total CD11b integrin on the cd14 -/- DCs and found overall a strong decrease in integrin activity, due to less active integrins on cd14 -/- DCs (line 249-251, Figure 3I and Supplementary Figure 3G). However, ON stimulation of cd14 -/- DCs still increased activity of CD11b integrins, pointing towards a CD14 independent mechanism of integrin hyperactivation after ON stimulation. We therefore revised this statement in the manuscript.

- It remains unclear whether interference with CD14 activity by UPEC ON has any impact in vivo. The study would be much improved if the authors could compare host survival and bacterial persistence upon infection of wt (and possibly CD14ko) mice with UPEC On and UPEC OFF.

Bacterial persistence assays with type 1 piliated and non-piliated UPEC were performed by others (Mulvey et al., Science, 1998; Mulvey, Schilling and Hultgren, Infection and Immunity 2001; Spaulding et al., Nature, 2017; Spaulding at al, *eLife*, 2018) and we think they are beyond the scope of our manuscript. We thus focused on the downstream effects of the FimH – CD14 interaction: we identified that UPEC ON mutants overactivate the NFAT pathway in DCs, leading to increased expression of immune-modulatory cytokines (line 316-325, Figure 4G, 4H and Supplementary Figure 4D). Additionally, we have proven that the observed effects also hold true in vivo (line 381-412, Figure 7). In line with secreted factors being affected, we found strong non-cell autonomous effects blurring the cellular phenotypes when differentially stimulated DCs were co-applied (Figure 7E and F).

The authors identify a potential binding site between FimH and CD14 and then convincingly show that these molecules physically interact. FimH therefore appears to be a critical determinant of the pathogenic effect of type 1 pili in DCs.- The interaction site between FimH and CD14 is said to occur "in an area not involved in LPS binding". It would be important if the authors supported this claim experimentally, by measuring whether FimH alters inflammatory gene expression and/or cytokine expression and in LPS-stimulated DCs. More generally, the study would greatly benefit it the authors could- CD14 acts in DCs in a ligand dose-dependent manner. Can the authors show DC responses to different concentration of UPEC ON/OFF?

We analyzed the engulfment of CD14 after UPEC ON and OFF stimulation, which was similar between the two stimuli (line 247-249, Figure 3H), and we analyzed cytokines expressed after UPEC ON and OFF stimulation, where we found no difference in the qualitative cytokines production usually linked to LPS stimulation of monocytes (IL-6, IL-1beta, INF-γ and TNF-α) (line 244-247, Figure 3G). We therefore confidently assume, that the FimH binding site on CD14 is indeed different enough from the LPS binding site and that LPS recognition is not inhibited by type 1 pili binding to CD14. We also checked if DCs react in a dose depended manner to UPEC ON and OFF mutants and found that DCs react with a gradual increase in adhesion the more ON mutants are added in a mixture of UPEC ON/OFF mutants (we did not add this data to the manuscript).

Reviewer #3:The authors explore the ability of the pili, a virulence factor typical of Enterobacteriaceae and of particular relevance in the study of uropathogenic *E. coli* (UPEC), to regulate the host immune response. Pili are particularly interesting as their expression can be turned on and off cyclically by bacteria due to phase variation, which can allow modulation of the virulence of isogenic bacteria. While it is known that pili modulate bacterial adhesion to host cells, little is known of the direct immunomodulatory properties of pili. Here the authors use a clever mutant *E. coli* model which is locked either in the ON (pilus expressor) or OFF( pilus non expressor) condition to study the ability of the pilus to influence dendritic cell (DCs) responses. The authors convincingly demonstrate that: (a) Pilus expressing (ON)-*E. coli* can inhibit the capacity of DCs to stimulate transgenic T cells in vitro, (b) ON-*E. coli* can down-modulate the ability of DCs to upregulate the costimulatory molecule CD40 and to a lesser extent CD80 and CD86, (c) ON-*E. coli* reduces DCs motility by enhancing the activation of beta2 integrins, which results in DCs being retained in the tissues and migrating less efficiently into the lymph, (d) CD14 is the receptor for pili proteins and is interaction between pili and CD14 is responsible for modulating some DCs responses.While the experiments are convincing and well controlled, the authors fail to evaluate key innate functions of dendritic cells, such as cytokine production, where CD14 plays an important regulatory role, either indirectly by regulating TLR internalization and activation of the endosomal pathway, or directly by activating calcium signaling and NFAT activation. It is also unclear whether CD14 binding is blocking its ability to respond to other ligands (i.e. LPS). This tampers the impact of the conclusions. Moreover the absence of any in vivo infection experiment, tampers the enthusiasm for the paper. It's difficult to interpret the claim that pili influence DCs ability to activate T cells with only limited in vitro experiments. DCs ability to influence adaptive immunity is dependent on several things, including antigen transport, cytokine production, cell-cell interaction, migration etc…that cannot be fully recapitulated in vitro. Also, DCs exert important innate immune functions locally in the tissues, that are not only the activation of T cells and do not require their migration into the lymph, an aspect that is not taken in consideration by the authors. In conclusion, while the experiments are sound and controlled, and the idea is of great general interest in the context of host pathogen interactions, the analyses are insufficient to understand the immunomodulatory properties of pili with a level of depth that can be meaningful in an infection context.

We thank reviewer 3 for the comments and criticism on our work and for appreciating our approach of phase-locked mutant generation. We fully agree that we did not show cytokine production as a key function of DCs, as also pointed out by reviewer 1 and 2. We therefore analyzed cytokine levels after UPEC ON and OFF stimulated DCs in a quantitative manner using a cytokine array assay to understand if production of pro-inflammatory cytokines and LPS-driven cytokines are altered after ON stimulation (line 244-247 and line 321-325, Figure 3G and 4H). We found immune-modulatory cytokines upregulated by ON mutants (line 321-325, Figure 4H), due to overactivation of the NFAT pathway (line 316-321, Figure 4G). However, LPS-driven cytokines were equally expressed, and therefore we confidently assume that binding of FimH to CD14 is not blocking binding and recognition of LPS (line 244-247, Figure 3G). Furthermore, this is supported by CD14 endocytosis being similar between ON and OFF stimulation (line 247-249, Figure 3H).

We agree that the lack of in vivo experiments undermined the relevance of our finding in the infection context. We therefore performed in vivo migration assays and in vivo T cell proliferation assays, as outlined in detail in the reply to reviewer 1 and 2 (data presented in a new figure, Figure 7). Briefly, we found migration of DCs into the lymph node reduced after ON stimulation, compared to OFF stimulation (line 396-402, Figure 7C). However, as soon as we mixed ON and OFF stimulated DCs before injection, we found same amount of DCs reaching the lymph node irrespective of the stimulus.

We assume that these effects are due to non-cell autonomous effects the DCs face once the DCs are in close contact with differentially stimulated DCs (Bardou et al., EMBOJ, 2021). T cell priming in vivo did not seem different after ON and OFF stimulation when looking at the overall amount of dividing T cells, however less T cells reached further rounds of division after ON stimulation reflected in the proliferation index (line 381-388, Figure 7A). We hypothesize that ON stimulation therefore also affects in vivo T cell priming, but given that inside the host antigen transfer to resident DCs in the lymph node occurs (Kleindienst and Brocker, The Journal of Immunology, 2003), we think that the effect is not as pronounced as in vitro. Given this non-cell autonomous effect, we are not convinced if infection models (for example bladder infection models of mice) would lead to additional insights for the interaction proposed of FimH-CD14. The identified interaction seems to be especially relevant at the individual cell-to-cell level and might be overlooked in the context of infection models.

As mentioned above, while the paper does a good job at identifying pili as structures able to influence DC responses through CD14, they fail to evaluate a number of important functions of dendritic cells that are directly regulated by the receptor CD14, such as cytokine production downstream of TLR4 and/or directly downstream to CD14. Moreover, they do not go into sufficient depth neither regarding the molecular mechanism downstream to CD14 that could explain the observed phenotypes (are pili stimulating responses downstream of CD14? Are they inhibiting LPS binding? Are they inhibiting interactions between CD14 and TLR4 or TLR2?) , nor the ability of pili to influence the overall outcome and/or the activation of DCs during an infection in vivo.The authors also discuss their results stressing that pili inhibit the capacity of DCs to activate T cells, however, only one initial experiment evaluating T cell activation is performed, in an in vitro setting with transgenic TCR bearing T cells. While the authors demonstrate that the time of contact between T cells and DCs is augmented in the presence of pilus expressing bacteria, they do not explore the consequences of this enhanced contact and the relative impact of integrin activation vs costimulatory molecules expression in influencing T cell responses. While the topic is of great potential interest and the authors have devised an interesting tool to specifically study the importance of pili, they need to add depth to their work in order to take complete advantage of the tool they created and to help to answer the question they have posed.Even taking into account the difficulty and the regulatory burden of animal experimentation, the authors could do a better job in dissecting the molecular mechanisms underlying the ability of pili to inhibit DCs and the dynamics of DCs-T cell interaction.

We now analyzed cytokine expression and identified the NFAT pathways to be upregulated upon UPEC ON stimulation, which leads to upregulation of NFAT-driven cytokines (GM-CSF, IL-4, IL-10; line 321325, Figure 4H). Blocking NFAT activation by calmodulin partially rescued the expression of costimulatory molecule expression (line 316-321, Figure 4G), and we therefore assume that additional pathways next to NFAT might be affected upon FimH binding to CD14. We also found LPS-driven cytokine expression (TNF-α, INF-γ, IL-6, IL-1beta) to be the same after ON and OFF stimulation, leading us to conclude that LPS binding/activation of CD14 is not inhibited by FimH binding (line 244-247, Figure 3G). This is further supported by the fact that CD14 still becomes internalized upon ON stimulation (line 247-249, Figure 3H).

Additionally, we also analyzed expression of co-stimulatory molecules on tlr4 -/- DCs (line 231-236, Figure 3C). In line with the increase in adhesion and the decrease in migration, tlr4 -/- DCs displayed decreased expression of co-stimulatory molecules after UPEC ON stimulation, as compared to UPEC OFF stimulation, although expression of co-stimulatory molecules was generally low.

To dissect if overactivation of integrins and the observed increased interaction time between DCs and T cells or if only downregulation of co-stimulatory molecules is causative for the dampened response of T cells, we performed in vitro T cell priming using beta2 -/- DC. We found that ON stimulated beta2 -/- DCs are still unable to efficiently prime T cells, compared to OFF stimulated beta2 -/- DCs (line 211216, Figure 2G and Supplementary Figure G). Thus, overactivation of integrins and the therefore increased DC-T cell interaction time is not causative for the lack of T cell response after UPEC ON stimulation; but what causes it, is the lack of co-stimulatory molecules. This is further supported by our finding, that cd14 -/- DCs still show overactive integrins after ON stimulation as compared to OFF stimulation (line 249-251, Figure 3I and Supplementary Figure 3G); although integrin activity was generally strongly reduced, when compared to WT DCs.

[Editors’ note: what follows is the authors’ response to the second round of review.]

Reviewer #1:The authors incorporated many requests previously made. In particular, they added new exciting results re NFAT relevance in their model and about cell-intrinsic and cell-extrinsic requirements. in vivo data, though only based on in vitro stimulated DCs, nicely complement the findings of the authors. The authors also analyzed additional isolates that suggest bacterium-specific behaviors.Although the authors did not perform in vivo infections, and some of their (new) findings open more mechanistic questions, the elegant ON/OFF model utilized, the identification of CD14 as a key regulator of responses to FimH, together with the new data provided make this new version of the manuscript exciting.I have only a few changes to request:1) Please properly cite in line 248, 249 the citation reported in line 1629.

As the results discussed previously in line 248-249, now line 251-255, are about CD14 internalization upon UPEC ON and OFF stimulation of DCs and thus are not a direct reference of the results reported by Zanoni et al. where they show that TLR4 endocytosis is controlled by CD14 in a LPS-dependent manner, we decided to cite Zanoni et al. in the Methods section for using their flow cytometry based method on reporting CD14 internalization (line 1166).

2) Re Figure 3G: Please, stress that RANTES is an ISG and, thus, the capacity of CD14 to drive interferon-dependent responses upon LPS encounter is not altered by the presence of FimH. Ideally, authors should have measured type I interferons to prove that CD14-intrinsic responses to LPS are not altered, but this point can be just mentioned, also based on data on CD14 internalization.

We now stressed that CCL5 is an Interferon-stimulated cytokine (line 247-248). Additionally, we assayed Type 1 interferon production as a means of Interferon-α 1 production. However, we did not find any production of Interferon-α 1 in our DCs, neither after UPEC ON, nor after UPEC OFF stimulation. Nevertheless, we included the results in line 252-254 and Figure 3—figure supplement 1H.

3) Always re Figure 3G: I would strongly encourage the authors to remove IFNγ from the cytokines measured. Murine DCs do not efficiently produce IFNγ in response to LPS/gram-negative bacteria. And indeed the authors measured extremely low levels, if any, of this cytokine.

We removed IFNγ from our figures, as indeed our murine-derived DCs did not produce it efficiently.

4) Please add to the discussion or at the end of the paragraph a brief line about the results shown in Figure 3I about the maintenance of increased integrin activity in CD14-/- cells exposed to ON bacteria.

We now rephrased the discussion in line 257-260 in order to better stress the maintenance of increased integrin activity: “Finally, although integrin activity was reduced in cd14-/- DCs (Figure 3I and Figure 3—figure supplement 1I), UPEC ON bacteria still slightly increased CD11b activity on cd14-/- cells when compared to UPEC OFF bacteria, showing that as opposed to downregulated co-stimulators, integrin activation did not strictly depend on the presence of CD14.”

Reviewer #2:The authors have performed a number of new experiments to address my initial concerns. The new manuscript shows that Pili act through the CD14-NFAT axis to induce the production of immunomodulatory cytokines that, in turn decrease upregulation of costimulatory molecules. The authors also added in vivo data modeling DCs migration by injecting OVA loaded-, *E. coli*-activated-, DCs in the footpad and analyzing migration and proliferation that demonstrate that the combination of defects in migration as well as costimulatory molecule affects T cell proliferation, albeit with a small size effect, and that the effect is not cell autonomous (consistently with the involvement of NFAT mediated cytokine production in the phenotype).The new data support idea that CD14 control of cytokine production through NFAT can induce the downregulation of costimulatory molecules, however I believe that some important controls should be included:The authors do not show any comparison between CD14 Ko DCs and WT DCs in term of cytokine production, and do not show whether CD14 KO and FK506 have the same effect on cytokine production and downregulation of costimulatory molecules.While it might be beyond the authors scope, it would be interesting to identify which NFAT dependent cytokine is mediating the phenotype observed.Moreover the small size of the in vivo effect, as well as the fact that it is not cell autonomous somewhat lower the enthusiasm for the conclusions, in particular, due to the fact that pili expression can be turned on and off in vivo, thus mimicking the effect of a mixed stimulation. Importantly assessment of mixed transfer on T cell proliferation is missing!

We now included a side-by-side comparison of cytokine production of WT DCs, WT DCs pretreated with FK506 inhibitor and *cd14^-/-^* DCs (line 332-339, Figure 4H). We compared NFAT-dependent cytokine production after ON and OFF stimulation, as we found different baseline expression of cytokines especially after FK506 treatment. We found that overproduction of NFAT-dependent cytokines is dependent on CD14 and the NFAT pathway, as deletion of *cd14* or inhibition of calcineurin by FK506, lead to similar production of cytokines after ON and OFF stimulation.

Regarding the in vivo relevance, we very much hope that infectiologists proficient in performing adequate infection experiments, will follow up on our mechanistic findings. It is beyond the scope of our work and beyond our core expertise to carry out mouse infection experiments, that (in order to fulfill highest standards) will have to include a thorough baseline characterization of both the innate and adaptive responses to ON and OFF bacteria as well as specific interference with the FimH-CD14 interaction on the host as well as the pathogen side.

Some other concerns include:a) WT, CD14 and TLR4 KO cells are never depicted in the same graph and therefore I deduce that the experiments on each cell genotype are independent experiments. These comparisons should be done in the same experiment to be able to compare KO with WT. Indeed the author state in line: 232-234 " It was previously shown that tlr4-/- DCs express reduced levels of co-stimulatory molecules (Shen et al., 2008) and this was also true for our in vitro generated tlr4-/- DCs". However, TLR4 -/- DCs are never compared to WT DCs in the manuscript.

We do not compare WT and mutant DCs directly, as usually the mutant DCs have a different baseline expression of co-stimulatory molecules and of other analyzed genes. For the *tlr4^-/-^* DCs we were particularly referring to expression of co-stimulatory molecules in comparison to the isotype control (see Figure 3—figure supplement 1C, dashed lines). In order to avoid any confusion, we now changed the sentence in line 232-234 to “It was previously shown that *tlr4^-/-^* DCs express reduced levels of co-stimulatory molecules (Shen et al., 2008) and this was also true for our in vitro generated *tlr4^-/-^* DCs as they express hardly any CD40 when compared to the isotype control level (Figure 3—figure supplement 1C).” for easier understanding what we compare the reduced levels to.

b) While internalization of CD14 Is assessed, production of type I interferons is never assessed and should be considered as CD14 trafficking of TLR4 can influence IFN production.

We now assayed Interferon-α 1 and found no production of Interferon-α 1 in our DCs, neither after UPEC ON nor after UPEC OFF stimulation. We included the results in line 252-254 and Figure 3—figure supplement 1H.

c) While CD80 and CD86 expression are clearly depending on CD14, CD40 expression seems to be TLR4 dependent. The author should discuss this point.

Although from the bar plots in Figure 3C and 3F one could draw the conclusion that expression of CD40 after UPEC ON and OFF stimulation is TLR4-dependent, the histograms in Figure 3—figure supplement 1 C and F show that *tlr4^-/-^* DCs barely express any CD40 over the isotype background level. We therefore would not conclude that CD40 expression in our infection model is TLR4-dependent. We now state that CD40 expression is slightly affected in *cd14^-/-^* DCs after UPEC ON stimulation, compared to OFF stimulation in line 244-246.

d) In line 244 LPS driven cytokines is incorrect. TLR4 driven or NfKB dependent cytokines would be more rigorous.

We changed LPS-driven cytokines to TLR4-dependent throughout the text.